# OPMapper: Enhancing Open-Vocabulary Semantic Segmentation with Multi-Guidance Information

**Xuehui Wang**[1*]    **Chongjie Si**[1*]    **Xue Yang**[2]    **Yuzhi Zhao**[3]    **Wenhai Wang**[4]
**Xiaokang Yang**[1]    **Wei Shen**[1✉]

[1]MoE Key Lab of Artificial Intelligence, AI Institute, School of Computer Science, SJTU
[2]School of Automation and Intelligent Sensing, SJTU
[3]City University of Hong Kong    [4]MMLab, Chinese University of Hong Kong

{wangxuehui,wei.shen}@sjtu.edu.cn

## Abstract

Open-vocabulary semantic segmentation assigns every pixel a label drawn from an open-ended, text-defined space. Vision–language models such as CLIP excel at zero-shot recognition, yet their image-level pre-training hinders dense prediction. Current approaches either fine-tune CLIP—at high computational cost—or adopt training-free attention refinements that favor local smoothness while overlooking global semantics. In this paper, we present OPMapper, a lightweight, plug-and-play module that injects both local compactness and global connectivity into attention maps of CLIP. It combines Context-aware Attention Injection, which embeds spatial and semantic correlations, and Semantic Attention Alignment, which iteratively aligns the enriched weights with textual prompts. By jointly modeling token dependencies and leveraging textual guidance, OPMapper enhances visual understanding. OPMapper is highly flexible and can be seamlessly integrated into both training-based and training-free paradigms with minimal computational overhead. Extensive experiments demonstrate its effectiveness, yielding significant improvements across 8 open-vocabulary segmentation benchmarks.

## 1 Introduction

Open-vocabulary semantic segmentation (OVSS) aims to assign every pixel in an image to *any* category rather than to a fixed, closed set of labels. In real-world applications, exhaustively annotating all possible classes is infeasible. Hence, OVSS has become a touch-stone for measuring a model's ability to generalize beyond its training categories. Large-scale vision–language pre-trained models (VLMs) [35, 1, 7, 38], epitomised by Contrastive Language-Image Pre-training (CLIP) [40], have unlocked promising avenues for OVSS [14, 20, 51, 33, 48, 50, 52]. However, leveraging CLIP for dense prediction is far from straightforward. Because CLIP is optimised under image-level supervision, its training paradigm is intrinsically misaligned with the pixel-wise granularity demanded by open-vocabulary detection [21, 54, 30, 31] and segmentation [28, 15, 45]. Consequently, naïve adaptations typically yield noisy, spatially inconsistent masks and unsatisfactory performance, underscoring the need for principled strategies that bridge the supervision gap between image-level pre-training and pixel-level inference.

A seemingly direct remedy is to finetune the entire CLIP backbone or the mask generator [10, 49], as shown in Figure 1(a)(c). Although this strategy can inject task-specific inductive bias, it comes at a steep cost: (i) it demands large-scale pixel-level annotations and significant GPU hours; (ii) gradient updates inevitably distort CLIP's carefully calibrated image–text embedding space, diminishing the

---

✉ Corresponding author.

39th Conference on Neural Information Processing Systems (NeurIPS 2025).

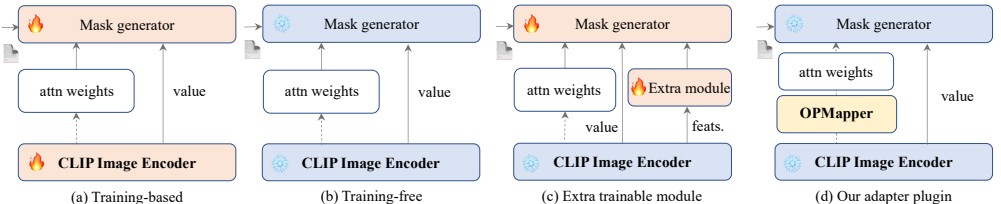

Figure 1: Different approaches for achieving open-set segmentation using CLIP. (a) illustrates the training-based paradigm, where all or part of CLIP's and mask generator's parameters are finetuned. (b) demonstrates the training-free manner, where the key idea is directly enhancing the query-key attention to boost performance. (c) illustrates a scheme where an additional trainable module is inserted and jointly trained with the mask generator to achieve better performance. (d) highlights the versatility of our mapper, which are trained offline and can be applied across (a), (b), and (c), or their hybrid paradigms. For simplicity, we illustrate its application only in training-free methods here.

open-vocabulary generalization it was prized for. To sidestep these drawbacks, a parallel line of *training-free* methods, as shown in Figure 1(b), has emerged [45, 28, 15]. Their key insight is to re-engineer the self-attention computation rather than alter model parameters. Concretely, they impose matrix transformations on the query–key similarity matrix—e.g., introducing spatial priors—to craft task-aware attention weights, which in turn remix the unmodified value embeddings into pixel-discriminative features. Because the underlying value embeddings (and all learned parameters) remain unchanged—only their aggregation is reweighted—the original image–text alignment is preserved, and no additional training or labelled data are required [28, 15, 45]. Nevertheless, most existing designs hinge on the heuristic of *locality compactness*—assuming that neighbouring tokens possess the strongest semantic affinity—while undervaluing *global connectivity*, the broader relational dependencies among distant tokens that underpin high-level semantic structures. This limitation compromises a holistic understanding of image context; for example, recognizing a small patch of knitted fabric alone may be insufficient to distinguish whether it belongs to a coat or a towel.

The foregoing analysis naturally prompts a question: *Can we avoid redesigning the entire attention pipeline or updating millions of parameters?* In this paper, we answer this question by proposing **OPMapper** (Object-to-Pixel Mapper) that re-calibrates only the frozen last-layer query–key pair to produce *dense-aware* attention weights, as shown in Figure 1(d). This mapper shifts CLIP's focus from objects to pixels while balancing local compactness and global connectivity. To construct the OPMapper, we design one operative module and one training-only auxiliary module: Context-aware Attention Injection (CAI), which processes the query and key from the frozen last-layer of CLIP and injects local and global cues into them to predict attention weights; Semantic Attention Alignment (SAA), which nudges the enriched attention produced by CAI to stay aligned with textual prompts, allowing effective use of textual information to refine the representation of attention. Specifically, in the CAI module, we manually design an idealized set of attention weights guided by two principles: (1) tokens in close proximity receive higher attention, reinforcing local compactness; (2) tokens within the same semantic category also receive higher attention, promoting global connectivity. Aligning the predicted attention weights from CAI with this prior distribution effectively integrates both local and global dependencies into the attention mechanism. The SAA module, implemented as an iterative cross-modal fusion network, refines these CAI-enhanced attention weights by aligning them with textual features, thus facilitating a better text-image alignment. Once training is complete, the SAA module can be discarded, leaving CAI as the sole component of OPMapper.

Since OPMapper can convert CLIP's native outputs into dense-aware representations without altering parameters of the CLIP, it is highly flexible and can be seamlessly integrated into existing methods with minimal computational overhead, as shown in Figure 1(d), serving as a modular enhancement to improve both training-based/free approaches. Extensive experiments demonstrate its ability to significantly enhance the performance of existing baselines. Our contributions are as follows:

- We design a lightweight OPMapper that transforms CLIP's object-level attention into pixel-level attention, without modifying any of its parameters.

- To train the mapper, we devise two novel modules that inject both local compactness and global connectivity while preserving CLIP's inherent cross-modal alignment.

- Our OPMapper was integrated into 11 CLIP-based models and consistently delivered significant performance improvements across 8 benchmarks.

## 2 Related works

### 2.1 Vision-language foundation model

Vision-language models [40, 35, 1, 7, 38] have greatly advanced the ability to address open-set visual tasks. As a representative contrastive learning-based approach, CLIP [40] is widely employed in open-set recognition for its effective alignment of visual and linguistic representations. CLIP establishes the relationship between vision and language by leveraging large-scale image-text pairs, resulting in two parallel encoders: one for encoding textual descriptions into text embeddings and the other for encoding images into context-aware global visual embeddings, thus enabling zero-shot recognition capabilities for various visual tasks.

### 2.2 Open vocabulary semantic segmentation

Open-vocabulary semantic segmentation [28, 15, 45, 14, 20, 29], also referred to as zero-shot semantic segmentation, aims to segment images into arbitrary categories defined by textual descriptions. This task has gained significant attention due to its potential to generalize beyond fixed class sets, leveraging the capabilities of VLMs such as CLIP. However, the pretraining objective of CLIP, which focuses on image-level alignment between visual and textual features, makes it inherently unsuitable for dense prediction tasks [14, 20, 33, 48, 50, 52]. To address this limitation, various methods have been proposed to extend CLIP for dense segmentation, which can be broadly categorized into training-based approaches and training-free approaches

Training-based methods typically involve optimizing all parameters of CLIP. CAT-Seg [9] and SED [46] employ cost aggregation to refine the pixel-text cost volume and generate per-pixel classification predictions. SAN [49] enhances CLIP with an adapter, enabling both mask proposal generation and recognition. Meanwhile, FC-CLIP [53] leverages CLIP as a shared backbone for unified mask generation and classification. Mask-Adapter [32] extracts semantic activation maps from proposal masks, enriching contextual information and enhancing the alignment. SCAN [17] integrates CLIP's generalized semantic prior into the proposal embedding, preventing it from collapsing onto known categories. While these approaches achieve promising results, the substantial computational resources required for CLIP training pose significant challenges for practical deployment.

Training-free methods effectively address the aforementioned resource constraints by keeping CLIP's parameters fixed. Instead of fine-tuning the model, these approaches focus on designing more effective attention weights—specifically, optimizing the interaction between queries and keys within the attention mechanism to guide the aggregation of value tokens, thereby enhancing the final output. Specifically, SCLIP [45] reformulates the $q$-$k$ attention weight as a combination of $q$-$q$ weight and $k$-$k$ weight to better capture local semantic coherence. ProxyCLIP [28] leverages the output features from other vision foundation models, such as DINO [3] and MAE [23], to generate attention weightss. MaskCLIP [15], on the other hand, enforces the $q$-$k$ attention weights to be an identity matrix, thereby directly guiding visual feature aggregation. However, as discussed in the introduction, these methods predominantly focus on token-level local compactness while overlooking global connectivity, leading to an incomplete understanding of the image. To address this limitation, in the following section, we introduce a novel approach that effectively leverages both local compactness and global connectivity, enhancing the overall comprehension of visual content.

## 3 Preliminary on CLIP

The CLIP model comprises an image encoder, denoted as $\texttt{Enc}_v$, and a text encoder, denoted as $\texttt{Enc}_t$. Given an input image $\mathbf{I}$ and a series of textual descriptions $\mathbf{D} = \{\texttt{a photo of a \{class\}}\ \} \in \mathbb{R}^L$, where $\texttt{class}$ represents the categories present in the image and $L$ is the number of classes, the visual feature $\mathbf{F} \in \mathbb{R}^{H \times W \times C}$ and the text embeddings $\mathbf{T} \in \mathbb{R}^{L \times C}$ are obtained as follows:

$$\texttt{CLS}, \mathbf{F} = \texttt{Enc}_v(\mathbf{I}), \quad \mathbf{T} = \texttt{Enc}_t(\mathbf{D}), \tag{1}$$

where $\texttt{CLS} \in \mathbb{R}^C$ represents the class token with $C$ representing the channel. It aggregates global contextual information and is used for classifying the entire image.

For semantic segmentation, given an input image, we first utilize CLIP to extract the corresponding text embedding and the *query*, *key*, and *value* features from the final layer of its visual encoder, then process the *query* and *key* to generate a $q$-$k$ attention weights, which combined with the *value* to produce the final visual features. The output of the final layer of $\texttt{Enc}_v$ is employed to perform

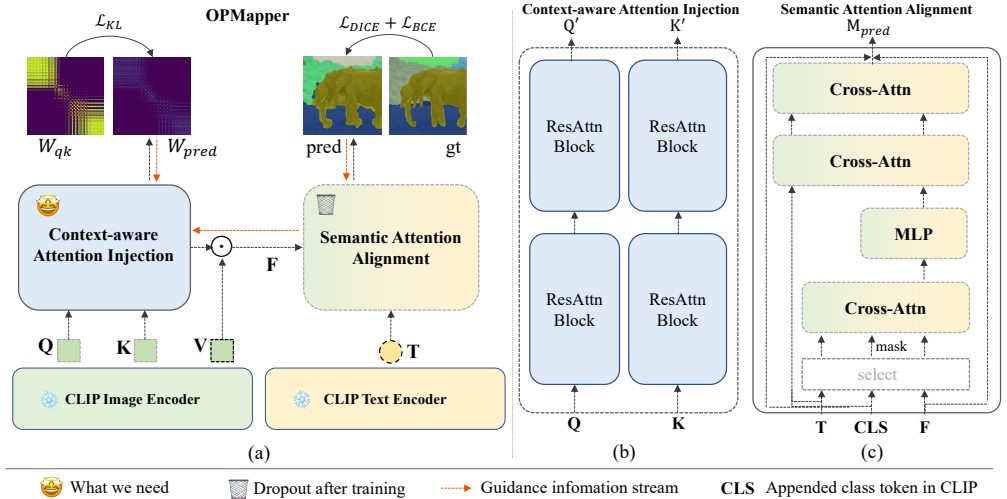

Figure 2: **Overall framework of our proposed model.** (a) illustrates the architecture of our OPMapper, which consists of two modules. (b) depicts the Context-aware Attention Injection (CAI) module with detailed network structure provided in the supplementary materials. (c) represents our Semantic Attention Alignment (SAA) module, where the yellow-green gradient indicates the fusion of visual and textual information. Note that after training, the SAA module will be dropout.

element-wise classification. This is achieved by leveraging the given text embedding $\mathbf{T}$ to generate a segmentation mask $\mathbf{M} \in \mathbb{R}^{H \times W \times L}$:

$$\mathbf{M} = h(\mathbf{F}\mathbf{T}^T) = h(\texttt{Attn}(\mathbf{Q}, \mathbf{K}, \mathbf{V})\mathbf{T}^T), \tag{2}$$

where $h$ is the argmax function and $\texttt{Attn}$ is the attention mechanism. $\mathbf{Q}$, $\mathbf{K}$, and $\mathbf{V}$ denote the $query$, $key$, and $value$, respectively. These representations are obtained by applying linear mappings to the previous layer's outputs, transforming input features for the attention mechanism. The text embedding $\mathbf{T}$ is typically kept fixed during training to preserve their semantic richness, effectively functioning as a comprehensive set of classifiers.

## 4 Methodology

As illustrated in Figure 2, OPMapper consists of two key modules: (1) Context-aware Attention Injection (CAI), which injects an idealized attention weight distribution that encodes both locality and global semantic coherence, effectively guiding the attention mechanism to capture meaningful token interactions. CAI is the sole element appended to the frozen CLIP stream, thus constituting the entire OPMapper proposed in this work. (2) The auxiliary Semantic Attention Alignment (SAA) module, which further refines the enriched attention by aligning it with textual prompts through an iterative fusion strategy, ensuring a more comprehensive integration of visual and textual information. We will elaborate the overall loss functions that jointly optimize these modules and details about integration with other methods in Sec 4.3.

### 4.1 Context-aware attention injection

#### 4.1.1 Manually designed attention weights

Firstly, local compactness ensures that features at adjacent spatial locations exhibit smooth transitions, which is crucial for fine-grained tasks such as object boundary delineation. Meanwhile, global connectivity captures long-range semantic relationships, enabling the model to recognize holistic object structures even when parts of an object are spatially distant. We argue that a prior attention weight matrix should simultaneously account for both local compactness and global connectivity to achieve a well-balanced representation. However, in a training-free setting, it is nearly impossible for the model to naturally incorporate both aspects using only existing supervision signals and standard computational mechanisms. To address this limitation, we introduce an idealized attention weight matrix that explicitly encodes both local and global relationships. This matrix serves as a guiding

reference for the attention mechanism, providing an inductive bias that encourages a more structured and semantically coherent attention distribution.

Given a ground truth label map $\mathbf{L} \in \mathbb{R}^{H \times W}$, where each pixel is assigned to one of $C$ semantic classes, we first down-sample it to align with the output resolution of the CLIP image encoder:

$$\hat{\mathbf{L}} = f_d(L, \alpha) \in \mathbb{R}^{\hat{H} \times \hat{W}}, \tag{3}$$

where $\alpha$ is the scaling factor, ensuring that the segmentation label map is compatible with the feature grid of CLIP. $f_d$ represents the down-sampling function, and $\hat{H}, \hat{W} = \frac{H}{\alpha}, \frac{W}{\alpha}$ represent the size of the down-sampled label map. Next, we construct the prior attention weights based on two key principles: (1) spatially adjacent tokens should receive higher attention scores to preserve local compactness, ensuring smooth feature transitions, and (2) tokens that share the same semantic category should be more strongly connected, reinforcing global connectivity and capturing long-range dependencies.

**Local Compactness.** To preserve local compactness, we define the local attention matrix $\mathbf{W}_l \in \mathbb{R}^{\hat{H}\hat{W} \times \hat{H}\hat{W}}$, which assigns higher scores to spatially closer tokens within the same semantic category. The attention score between two positions $i$ and $j$ is computed as:

$$\mathbf{W}_l[i,j] = \begin{cases} \exp(-d(i,j)), & \text{if } \hat{L}_i = \hat{L}_j, i \neq j \\ 0, & \text{otherwise.} \end{cases} \tag{4}$$

where $d(i,j)$ represents the Euclidean distance between positions $i$ and $j$ in the 2D spatial domain. This ensures that nearby tokens receive higher attention weights while distant tokens within the same class still maintain a meaningful connection. To prevent self-loops, diagonal elements (i.e., $i = j$) are set to zero. The matrix is then row-wise normalized using a softmax function, similar to $q$-$k$ attention computation in standard self-attention mechanisms.

**Global Connectivity.** To capture global connectivity, we construct a global attention matrix $\mathbf{W}_g \in \mathbb{R}^{\hat{H}\hat{W} \times \hat{H}\hat{W}}$, where each entry $(i,j)$ represents the semantic similarity between two spatial positions:

$$\mathbf{W}_g[i,j] = \begin{cases} 1, & \text{if } \hat{L}_i = \hat{L}_j, \\ 0, & \text{otherwise.} \end{cases} \tag{5}$$

This binary mask ensures that only tokens belonging to the same semantic category are connected, even they are far from each other. We then normalize each row by a mean operation to obtain a probabilistic attention distribution, giving all related tokens an equal share of attention.

### 4.1.2 Unified attention weight construction

To achieve a balanced integration of local and global dependencies, we combine the two attention matrices into a unified attention weight matrix as shown in Figure 3:

$$\mathbf{W}_{qk} = \lambda \mathbf{W}_l + (1 - \lambda) \mathbf{W}_g, \tag{6}$$

where $\lambda$ is an empirically determined scaling factor (set to 0.3) to adjust the relative contribution of local and global attention components. This fused attention weight matrix serves as a reference target for OPMapper, guiding the model to learn an optimal attention distribution that balances spatial smoothness and semantic coherence.

### 4.1.3 Supervising attention learning

OPMapper is integrated into the $q$-$k$ attention computation of CLIP's vision encoder. Specifically, given the final layer's query $\mathbf{Q}$ and key $\mathbf{K}$, OPMapper transforms them into enhanced representations:

$$\mathbf{Q}' = \mathtt{Mapper}_q(\mathbf{Q}), \quad \mathbf{K}' = \mathtt{Mapper}_k(\mathbf{K}), \tag{7}$$

where $\mathtt{Mapper}_q$ and $\mathtt{Mapper}_k$ are two mapper modules, each consisting of two cross-attention blocks identical to those used in CLIP. The refined attention weights $\mathbf{W}_p = \mathbf{Q}'\mathbf{K}'^\mathsf{T}$ are then computed using these transformed queries and keys.

To ensure OPMapper effectively learns to align with the designed local-global balanced attention, we employ KL divergence loss between the predicted weights $\mathbf{W}_p$ and the reference weights $\mathbf{W}_{qk}$:

$$\mathcal{L}_{\mathrm{KL}} = \sum_{i,j} \mathbf{W}_p[i,j] \log \frac{\mathbf{W}_p[i,j]}{\mathbf{W}_{qk}[i,j]}. \tag{8}$$

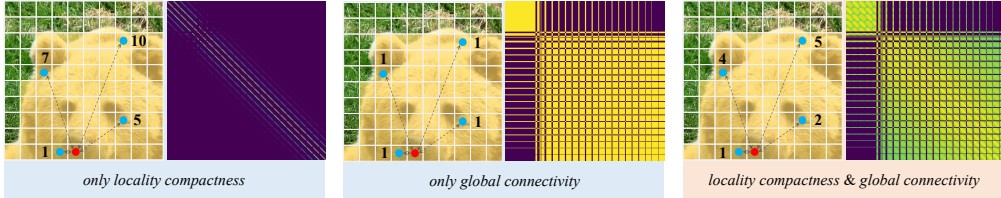

| | | |
|---|---|---|
| *only locality compactness* | *only global connectivity* | *locality compactness & global connectivity* |

Figure 3: Illustration of attention map construction. The left column shows how attention weights are assigned between patches, while the right column visualizes the resulting attention map. Red dots mark the selected tokens, and blue dots indicate the tokens whose similarity is computed. For simplicity, distances are represented by numerical values, with smaller distances implying higher weights. The example uses the bear category, visualizing attention relationships among all tokens.

By minimizing this loss, OPMapper learns to adjust the attention mechanism to better integrate local compactness and global connectivity, ultimately enhancing segmentation performance while maintaining computational efficiency.

## 4.2 Semantic attention alignment

While the CAI module injects local and global token relationships into the attention mechanism, it does not explicitly ensure alignment with textual prompts, which are crucial for open-vocabulary segmentation. Without proper alignment, the refined attention may fail to fully leverage semantic cues from language supervision, limiting its effectiveness in distinguishing visually similar objects belonging to different categories. To address this, we introduce the Semantic Attention Alignment (SAA) module, which further refines the enriched attention representation by aligning it with textual features. By iteratively fusing visual attention with text embeddings, SAA enhances the semantic consistency of attention weights, ensuring that object regions are not only spatially and globally coherent but also correctly associated with the corresponding textual descriptions.

As shown in Figure 2(c), SAA operates in a three-stage process:

1. Text-Guided Visual Enhancement: Text embeddings refine visual attention to enhance semantic consistency.
2. Visual-Guided Text Refinement: The enriched visual features reciprocally influence textual embeddings for better alignment.
3. Cross-Modal Attention Refinement: A final fusion step iterates between the two modalities, ensuring deep semantic integration.

### 4.2.1 Text-guided visual enhancement

To inject textual cues into visual features, we use text embeddings as keys and values in a cross-attention layer, with visual features as queries. This enables the model to reweight visual features based on their semantic proximity to textual categories. The operation is formulated as

$$\mathbf{F}_1 = \text{CA}(q = \mathbf{F}, k = \mathbf{T}, v = \mathbf{T}) \in \mathbb{R}^{\hat{H}\hat{W} \times C}, \tag{9}$$

where $\mathbf{F}_1$ represents the text-refined visual features, $\mathbf{F}$ represents the visual representation obtained using $\mathbf{W}_p$ as the attention weights, i.e., the result of applying this attention to the value embeddings. $\mathbf{T}$ denotes the feature generated by the text encoder from the encoded text prompt, and CA represents the cross attention mechanisms.

To further suppress the influence of irrelevant categories, we compute the cosine similarity between the CLS token and text embeddings, and between visual features and text embeddings, determining the relevance of each visual token to different categories:

$$\mathbf{S}_{\text{CLS}-\mathbf{T}} = \text{CosSim}(\text{CLS}, \mathbf{T}) \in \mathbb{R}^L,$$
$$\mathbf{S}_{\mathbf{V}-\mathbf{T}} = \text{CosSim}(\mathbf{F_1}, \mathbf{T}) \in \mathbb{R}^{\hat{H}\hat{W} \times L}. \tag{10}$$

From these similarity scores, we select the top-k categories for each visual token by identifying the highest similarity values in $\mathbf{S}_{\text{CLS}-\mathbf{T}}$ and $\mathbf{S}_{\mathbf{V}-\mathbf{T}}$. The final category selection is obtained by taking the

union of the top-$k$ categories from both scores:

$$\mathcal{C}_{\text{selected}} = \text{Topk}(\mathbf{S}_{\text{CLS}-\mathbf{T}}) \cup \text{Topk}(\mathbf{S}_{\mathbf{V}-\mathbf{T}}). \tag{11}$$

Using this selected category set, we construct an attention mask $\mathbf{M}_{attn} \in \mathbb{R}^{\hat{H}\hat{W} \times L}$, where each column determines the valid textual categories for the corresponding visual token. Specifically, if the $j$-th category is present in $\mathcal{C}_{\text{selected}}$, we assign $\mathbf{M}[:, j] = 1$, otherwise, it remains zero. Finally, we apply the computed mask to restrict attention weights only to the selected textual categories:

$$\mathbf{F}_1 = \text{CA}(q = \mathbf{F}, k = \mathbf{T}, v = \mathbf{T}, mask = \mathbf{M}_{attn}). \tag{12}$$

This process ensures that visual features are selectively enriched with relevant textual context rather than being influenced by unrelated categories.

### 4.2.2 Visual-guided text refinement

To establish bidirectional alignment, the roles of visual and textual features are reversed. Instead of text refining vision, updated visual features $\mathbf{F}_1$ now serve as keys and values, while text embeddings act as queries. This enables the textual features to dynamically adjust based on the enriched visual information:

$$\mathbf{T}_1 = \text{CA}(q = \mathbf{T}, k = \mathbf{F}_1, v = \mathbf{F}_1). \tag{13}$$

It ensures that textual embeddings incorporate relevant spatial and semantic cues, reinforcing visual-text consistency.

### 4.2.3 Cross-modal attention refinement

Finally, we establish a cross-modal feedback loop, integrating the refined textual representation $\mathbf{T}_1$ back into the visual feature space through another cross-attention layer:

$$\mathbf{F}_2 = \text{CA}(q = \mathbf{F}_1, k = \mathbf{T}_1, v = \mathbf{T}_1). \tag{14}$$

The resulting feature $\mathbf{F}_2$ represents the final semantically-aligned visual representation, where each token effectively captures both local-global dependencies and textual associations. The final segmentation prediction is computed via:

$$\mathbf{M}_{pred} = \texttt{Sigmoid}(\mathbf{F}_2\mathbf{T}^T). \tag{15}$$

This iterative fusion strategy ensures that object regions are not only structurally coherent but also correctly mapped to textual descriptions, improving segmentation accuracy in an open-vocabulary setting. Since we continue to use the CLIP text encoder's original text embeddings $\mathbf{T}$ without incorporating the refined textual features $\mathbf{T}_1$ from the visual refinement process, SAA is only applied during training. During inference, the model relies solely on the CAI module, making OPMapper a lightweight, plug-and-play enhancement that imposes minimal computational overhead. This design ensures that OPMapper can be seamlessly integrated into existing segmentation frameworks.

### 4.3 Loss function and adaptation details

**Loss function.** In addition to $\mathcal{L}_{\text{KL}}$, the loss function for OPMapper combines the DICE loss, to address class imbalance, and the binary cross-entropy (BCE) loss, to improve foreground-background discrimination. The overall loss function is:

$$\mathcal{L} = w_{\text{KL}}\mathcal{L}_{\text{KL}} + w_{\text{DICE}}\mathcal{L}_{\text{DICE}} + w_{\text{BCE}}\mathcal{L}_{\text{BCE}}, \tag{16}$$

where $w_{\text{KL}}$, $w_{\text{DICE}}$, and $w_{\text{BCE}}$ are hyper-parameters set to balance the contributions of each term.

**Adaptation details.** Applying OPMapper to other methods is straightforward. Since the SAA module is discarded after training, OPMapper effectively reduces to the CAI module at inference time. To integrate it into other methods, we only need feed the output query and key embeddings from the final layer of the CLIP visual encoder of the method into OPMapper, which maps them to attention weights suitable for dense prediction tasks. These attention weights are then used in the original computation flow to interact with the value representations, yielding the final visual features.

Table 1: **Quantitative comparison on 8 benchmarks.** Gains over the base model are shown in green. "Avg." indicates the mean performance across 8 datasets. †: Results refined with a post-processing toolkit such as DenseCRF. ‡: Models trained from scratch under identical batch sizes for fairness. All implementations are based on official codes.

| Model | Image Encoder | VOC-21 | VOC-20 | Context-60 | Context-59 | Object-171 | Stuff-171 | ADE-150 | CityScapes | Avg. |
|---|---|---|---|---|---|---|---|---|---|---|
| ▼ *Training based* | | | | | | | | | | |
| GroupViT [47]* | VIT-B/16 | 52.3 | 74.1 | 22.4 | 23.4 | 24.3 | - | 10.6 | - | - |
| TCL [5]* | VIT-B/16 | 51.2 | 77.5 | 30.3 | 24.3 | 30.4 | - | 14.9 | - | - |
| ZegFormer [14]* | VIT-B/16 | 65.5 | 89.5 | - | 45.5 | - | - | 18.0 | - | - |
| OVSeg [34]* | VIT-B/16 | - | 92.6 | - | 53.3 | - | - | 24.8 | - | - |
| SAN [49]‡ | VIT-B/16 | - | 92.7 | - | 51.8 | - | 40.2 | 29.9 | - | - |
| +OPMapper‡ | VIT-B/16 | - | 93.2 (+0.5) | - | 52.7 (+0.9) | - | 41.3 (+1.1) | 30.1 (+0.2) | - | - |
| CAT-Seg [10]‡ | VIT-B/16 | 75.2 | 93.2 | - | 55.1 | - | 43.8 | 30.7 | - | - |
| + OPMapper‡ | VIT-B/16 | 74.8(-0.4) | 94.0 (+0.8) | - | 56.2 (+1.1) | - | 43.9 (+0.1) | 31.3 (+0.6) | - | - |
| SCAN [17]‡ | VIT-B/16 | - | 94.9 | - | 56.8 | - | 44.1 | 30.5 | - | - |
| +OPMapper‡ | VIT-B/16 | - | 96.0 (+1.1) | - | 58.3 (+1.5) | - | 44.8 (+0.7) | 31.0 (+0.5) | - | - |
| ▼ *Training free* | | | | | | | | | | |
| Vanilla CLIP [40] | VIT-B/16 | 16.4 | 41.9 | 8.4 | 9.2 | 5.6 | 4.4 | 2.9 | 5.0 | 11.7 |
| MaskCLIP [15] | VIT-B/16 | 38.8 | 74.9 | 23.6 | 26.4 | 20.6 | 16.4 | 9.8 | 12.6 | 27.89 |
| +OPMapper | VIT-B/16 | 50.7 (+11.9) | 79.1 (+4.2) | 32.5 (+8.9) | 35.8 (+9.4) | 31.8 (+11.2) | 23.2 (+6.8) | 16.8 (+7.0) | 31.5 (+18.9) | 37.68 (+9.79) |
| SCLIP [45] | VIT-B/16 | 52.5 | 78.2 | 30.4 | 35.5 | 33.2 | 23.6 | 16.8 | 31.0 | 37.65 |
| +OPMapper | VIT-B/16 | 56.2 (+3.7) | 84.1 (+5.9) | 35.2 (+4.8) | 38.8 (+5.6) | 36.8 (+3.6) | 25.9 (+2.3) | 18.3 (+1.5) | 33.6 (+2.6) | 41.11 (+3.46) |
| ClearCLIP [27] | VIT-B/16 | 51.9 | 80.9 | 32.4 | 35.9 | 33.2 | 23.9 | 16.7 | 30.0 | 38.11 |
| +OPMapper | VIT-B/16 | 55.6 (+3.7) | 84.7 (+3.8) | 34.8 (+2.4) | 38.6 (+2.7) | 36.5 (+3.3) | 25.9 (+2.0) | 18.1 (+1.4) | 32.9 (+2.9) | 40.89 (+2.78) |
| ProxyCLIP [28] | VIT-B/16 | 61.3 | 80.3 | 35.3 | 39.1 | 37.5 | 26.5 | 20.2 | 38.1 | 42.29 |
| +OPMapper | VIT-B/16 | 62.8 (+1.5) | 84.3 (+4.0) | 36.0 (+0.7) | 40.1 (+1.0) | 38.7 (+1.2) | 27.1 (+0.6) | 20.3 (+0.1) | 37.2 (-0.9) | 43.33 (+1.04) |
| LPOSS [42] | VIT-B/16 | 60.2 | 80.2 | 35.0 | 36.9 | 34.7 | 25.3 | 21.2 | 37.6 | 41.39 |
| +OPMapper | VIT-B/16 | 63.7 (+3.5) | 84.9 (+4.7) | 35.7 (+0.7) | 37.9 (+1.0) | 35.2 (+0.5) | 26.4 (+1.1) | 22.1 (+0.9) | 40.0 (+2.4) | 43.24 (+1.85) |
| CASS [26] | VIT-B/16 | 64.3 | 88.3 | 36.9 | 39.6 | 38.1 | 26.2 | 20.1 | 39.8 | 44.16 |
| +OPMapper | VIT-B/16 | 67.0 (+2.7) | 90.0 (+1.7) | 38.2 (+1.3) | 40.1 (+0.5) | 38.8 (+0.7) | 27.1 (+0.9) | 20.8 (+0.7) | 41.0 (+1.2) | 45.37 (+1.21) |
| Vanilla CLIP [40] | VIT-L/14 | 8.2 | 15.6 | 4.1 | 4.4 | 2.7 | 2.4 | 1.7 | 2.5 | 5.2 |
| CaR [43]† | VIT-L/14&ViT-B/16 | 67.6 | 91.4 | 30.5 | 39.5 | 36.6 | - | 17.7 | - | - |
| MaskCLIP [15] | VIT-L/14 | 41 | 65.1 | 24.5 | 26.5 | 26.4 | 17.6 | 15.1 | 21.2 | 29.68 |
| +OPMapper | VIT-L/14 | 50.9 (+9.9) | 81.9 (+16.8) | 30.6 (+6.1) | 33.7 (+7.2) | 33.5 (+7.1) | 22.3 (+4.7) | 18.3 (+3.2) | 29.7 (+8.5) | 37.61 (+7.94) |
| SCLIP [45] | VIT-L/14 | 47.4 | 79.3 | 27.8 | 30.6 | 30.1 | 20.5 | 15.6 | 27.8 | 34.89 |
| +OPMapper | VIT-L/14 | 50.4 (+3.0) | 81.6 (+2.3) | 29.6 (+1.8) | 32.6 (+2.0) | 33.5 (+3.4) | 21.8 (+1.3) | 16.5 (+0.9) | 30.3 (+2.5) | 37.03 (+2.14) |
| ClearCLIP [27] | VIT-L/14 | 46.1 | 80 | 26.8 | 29.6 | 30.1 | 19.9 | 15 | 27.9 | 34.43 |
| +OPMapper | VIT-L/14 | 49.2 (+3.1) | 81.5 (+1.5) | 28.7 (+1.9) | 31.7 (+2.1) | 33.1 (+3.0) | 21.3 (+1.4) | 16.1 (+1.1) | 29.8 (+1.9) | 36.43 (+2.00) |
| ProxyCLIP [28] | VIT-L/14 | 59.3 | 82.6 | 33.1 | 35.7 | 38.2 | 24.2 | 20.8 | 36.3 | 41.28 |
| +OPMapper | VIT-L/14 | 59.2(-0.1) | 84.4 (+1.8) | 33.7(+0.6) | 36.9 (+1.2) | 39.1 (+0.9) | 25.1 (+0.9) | 21.7 (+0.9) | 37.8 (+1.5) | 42.24(+0.96) |

Table 2: Ablation study on CAI and SAA.

| Method | ADE | City | Context59 | Object | VOC | Avg. |
|---|---|---|---|---|---|---|
| CLIP | 2.9 | 5.0 | 9.2 | 5.6 | 41.9 | 12.9 |
| w/o SAA. | 17.8 | 32.0 | 37.9 | 35.2 | 83.5 | 41.3 |
| w/o CAI. | 18.3 | 31.3 | 37.5 | 35 | 83.7 | 41.2 |
| OPMapper | 18.1 | 32.9 | 38.6 | 36.5 | 84.7 | 42.2 |

Table 3: Ablation study on the construction of local and global attention weights.

| Settings | ADE | City | Context-59 | Object | VOC | Avg. |
|---|---|---|---|---|---|---|
| w/o $\mathbf{W}_g$ | 17.6 | 30.9 | 36.9 | 32.2 | 79.6 | 39.4 |
| w/o $\mathbf{W}_l$ | 17.5 | 32.4 | 38.3 | 35.1 | 83.8 | 41.4 |
| OPMapper | 18.1 | 32.9 | 38.6 | 36.5 | 84.7 | 42.2 |

## 5 Experiments

### 5.1 Datasets and baselines

We evaluate our method on 8 widely used semantic segmentation benchmarks, ensuring a comprehensive assessment. The datasets include PASCAL VOC20 (VOC21, with one additional background class) [18], PASCAL Context59 (Context60, with background) [39], COCO Object (Object) [2], COCO-Stuff (Stuff) [2], Cityscapes (City) [13], and ADE20K (ADE) [55]. Mean Intersection-over-Union (mIoU) is used as the evaluation metric across all datasets. Our mapper is trained on the COCO-Stuff dataset, which contains 118k densely annotated images spanning 171 categories. We then integrate our lightweight mapper into two types of existing methods: training-based methods, including SAN [49], SCAN [17] and CAT-Seg [10], and training-free methods, including Proxy-CLIP [28], ClearCLIP [27], SCLIP [45], LPOSS [42], CASS [26] and MaskCLIP [15]. We replace the standard attention operation with our mapper to construct the attention weights.

### 5.2 Implementation details

All experiments were conducted using MMSegmentation [11]. During training, COCO-Stuff images were resized to have a 384-pixel short edge, followed by proportional scaling and random cropping to $384 \times 384$. Data augmentation strategies aligned with CAT-Seg, including random flipping and patterned noise injection. We used the AdamW optimizer with learning rates of $5 \times 10^{-4}$ for the SAA and $5 \times 10^{-5}$ for the CAI. Our mapper is highly memory-efficient, requiring $\leq$ 2GB of GPU memory per batch, and was trained on four NVIDIA A100 GPUs (batch size: 8 per GPU) for 80,000 iterations,

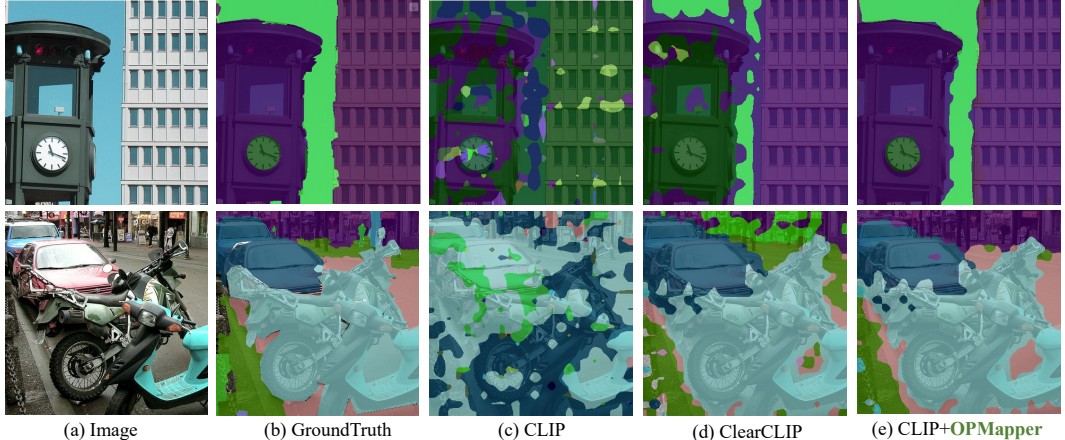

| (a) Image | (b) GroundTruth | (c) CLIP | (d) ClearCLIP | (e) CLIP+**OPMapper** |

Figure 4: Some visualization results evaluated on COCO-Stuff dataset. When equipped with our OPMapper, CLIP can obtain incredible performance. We provide more results in Appendix.

completing in 3–4 hours. Notably, OPMapper is trained entirely on the officially pre-trained CLIP model, keeping all CLIP parameters frozen. $w_{KL}$, $w_{DICE}$, and $w_{BCE}$ are set to 10, 5, 10, respectively. Further details are provided in the Appendix C.

## 5.3 Main results

The main results are presented in Table 1, demonstrating the effectiveness of our lightweight OPMapper across various benchmarks. In particular, when combined with CASS, OPMapper achieves superior performance across most benchmarks while substantially improving computational efficiency. By directly refining attention weights and integrating them with the feature maps from the vision-language model, OPMapper enhances segmentation quality in a training-free paradigm. Furthermore, when applied to trainable methods, OPMapper still yields notable gains, albeit with a less pronounced effect. This is likely because fine-tuning inherently optimizes attention weights for dense prediction, reducing the extent of additional improvements brought by our mapper. OPMapper consistently delivers significant performance improvements while maintaining exceptional efficiency, with a parameter size of only 0.8M. By eliminating redundant computations, OPMapper also facilitates faster inference, reinforcing its scalability and adaptability for diverse segmentation architectures.

To further illustrate OPMapper's effectiveness, we provide qualitative results in Figure 4, showcasing its ability to enhance segmentation through improved attention alignment. More experimental results can be referred to Appendix A.

## 5.4 Further analysis

To efficiently validate the effectiveness of each component, we conduct a series of ablation studies, primarily employing CLIP with the ViT-B/16 image encoder as the baseline, unless otherwise specified. This choice ensures that the evaluation remains unbiased and is unaffected by potential performance enhancements introduced by other CLIP variants.

Due to space limitations, we sincerely encourage readers to refer to the Appendix A / B for more detailed analyses, including more ablation studies, the visualization of the learned attention weights, the motivation behind our design, the evidence for our claim.

**Ablation on different modules** Table 2 present the ablation study of OPMapper. It is obvious that both CAI and SAA demonstrate substantial improvements over the baseline, with the average performance improving from 12.92 mIoU (baseline) to 41.28 mIoU (CAI) and 41.16 mIoU (SAA), respectively. Finally, by combining both modules, the average performance is further enhanced to 42.16 mIoU. These results confirm the complementary nature of the two modules.

**Ablation on local and global attention weights** To validate the effectiveness of the Local and Global Attention Weights ($\mathbf{W}_l$ and $\mathbf{W}_g$) constructed in the CAI module, we conduct ablation experiments where each component is used independently. The results, presented in Table 3, clearly demonstrate that the most substantial performance gains are achieved when both global and local attention weights

are jointly considered. This highlights the necessity of balancing local and global dependencies for optimal segmentation performance. To validate the rationale behind our attention weights design, we visualize the predicted attention weights and compare them with those from fully trained models in the supplementary materials. The results indicate that the attention weights generated by OPMapper closely resemble those from fully trained models, highlighting its effectiveness.

**Ablation on mixing coefficient ($\lambda$).** We conduct an ablation study on the choice of $\lambda$ in Equation 6 of our paper, which serves to balance the relative contributions of local compactness and global connectivity when manually constructing the prior weight map. As shown in Table 4, the best performance is achieved when $\lambda = 0.3$. As $\lambda$ increases beyond this point,

Table 4: Ablation on mixing coefficient ($\lambda$).

| Settings | ADE | City | Context59 | Object | VOC | Avg. |
|---|---|---|---|---|---|---|
| $\lambda = 0.1$ | 17.8 | 32.5 | 38.2 | 35.9 | 83.8 | 41.6 |
| $\lambda = 0.3$ | 18.1 | 32.9 | 38.6 | 36.5 | 84.7 | 42.2 |
| $\lambda = 0.5$ | 18.0 | 32.6 | 38.1 | 36.0 | 83.7 | 41.7 |
| $\lambda = 0.7$ | 17.7 | 32.3 | 38.0 | 36.0 | 83.4 | 41.5 |
| $\lambda = 0.9$ | 17.0 | 31.7 | 37.3 | 35.6 | 82.6 | 40.8 |

performance gradually deteriorates, suggesting that the current fusion strategy between local and global cues is optimal for generating pixel-level weight maps. In contrast, relying solely on either local compactness or global connectivity leads to a noticeable drop in performance.

**Ablation on the weight of three loss functions.** To investigate the impact of the three losses used during OPMapper training—KL/DICE/BCE losses—we explore various weight combinations within a predefined range of 5 to 10, chosen to match the learning rate of the whole model. As shown in Table 5, the best performance is achieved when the weights are set to $w_{KL} = 10$, $w_{DICE} = 5$, and $w_{BCE} = 10$. Reducing the weights of both $w_{KL}$ and $w_{BCE}$ to 5 results in a noticeable performance drop. Interestingly, increasing the $w_{DICE}$ to 10 also leads to a decline in performance, suggesting that an overemphasis on DICE loss may amplify its role in handling

Table 5: Ablation on the weight of three loss functions. Here, the setting (5, 5, 5) indicates that the $w_{KL}$, $w_{DICE}$ and $w_{BCE}$ are 5, 5, and 5, respectively. Other settings follow the same convention.

| Settings | ADE | City | Context59 | Object | VOC | Avg. |
|---|---|---|---|---|---|---|
| (5,5,5) | 16.2 | 30.7 | 36.2 | 34.5 | 81.6 | 39.8 |
| (5,5,10) | 17.4 | 31.7 | 38.1 | 35.7 | 83.5 | 41.3 |
| (5,10,5) | 15.7 | 30.2 | 35.4 | 34.1 | 80.9 | 39.3 |
| (5,10,10) | 16.7 | 31.3 | 37.6 | 35.3 | 83.3 | 40.8 |
| (10,5,5) | 17.7 | 31.9 | 38.4 | 35.9 | 83.9 | 41.6 |
| (10,5,10) | 18.1 | 32.9 | 38.6 | 36.5 | 84.7 | 42.2 |
| (10,10,5) | 16.5 | 31.1 | 38.0 | 35.2 | 82.6 | 40.7 |
| (10,10,10) | 17.6 | 31.7 | 38.1 | 35.7 | 83.5 | 41.3 |

class imbalance to an extent that disrupts the CAI module's ability to properly transform the object-aware query and key embeddings from the final layer of the CLIP's vision encoder into pixel-aware.

## 6 Conclusion

In this paper, we introduced OPMapper, a lightweight and flexible plug-and-play module designed to enhance open-vocabulary segmentation by integrating local compactness and global connectivity into the attention mechanism. OPMapper introduces Context-aware Attention Injection (CAI) to explicitly encode both spatial proximity and semantic coherence, and Semantic Attention Alignment (SAA) to iteratively refine attention by aligning it with textual features. Our approach is computationally efficient and requires no additional training of CLIP parameters, making it seamlessly compatible with both training-based and training-free paradigms. Extensive experiments demonstrate that OPMapper significantly improves segmentation performance across multiple benchmarks while maintaining minimal computational overhead. By effectively balancing local and global information within the attention mechanism, OPMapper offers a scalable and adaptable solution for advancing open-vocabulary segmentation.

**Limitations.** OPMapper's benefit diminishes when the CLIP vision encoder has been heavily fine-tuned (its parameter distribution drifts markedly from the original model). Currently, OPMapper excels only when this shift is moderate. We think (i) quantify the distributional gap between the fine-tuned and base encoders, then correct it to better accommodate OPMapper, and (ii) explicitly tie OPMapper's parameters to the statistics of the fine-tuned model are promising directions.

**Broader impacts.** Our method targets object segmentation beyond the training distribution, enabling more generalizable perception in robotics, autonomous driving, and related domains, ultimately advancing socially beneficial technologies.

## Acknowledgments

This work was supported by the NSFC under Grant 62322604, 62176159, Natural Science Foundation of Shanghai (25ZR1402268), and in part by the Shanghai Municipal Science and Technology Major Project under Grant 2021SHZDZX0102.

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

# Technical Appendices

## A   More experiments

### A.1   Qualitative results

As shown in Figure 5, we provide additional segmentation results for different datasets, including ADE20K [56], Cityscapes [12], VOC [18] as well as Pascal Context [19]. As illustrated in the Figure 5, inserting our OPMapper into ClearCLIP[27] leads to more coherent segmentation results by connecting semantically related regions and reducing fragmented or noisy mis-segmentations.

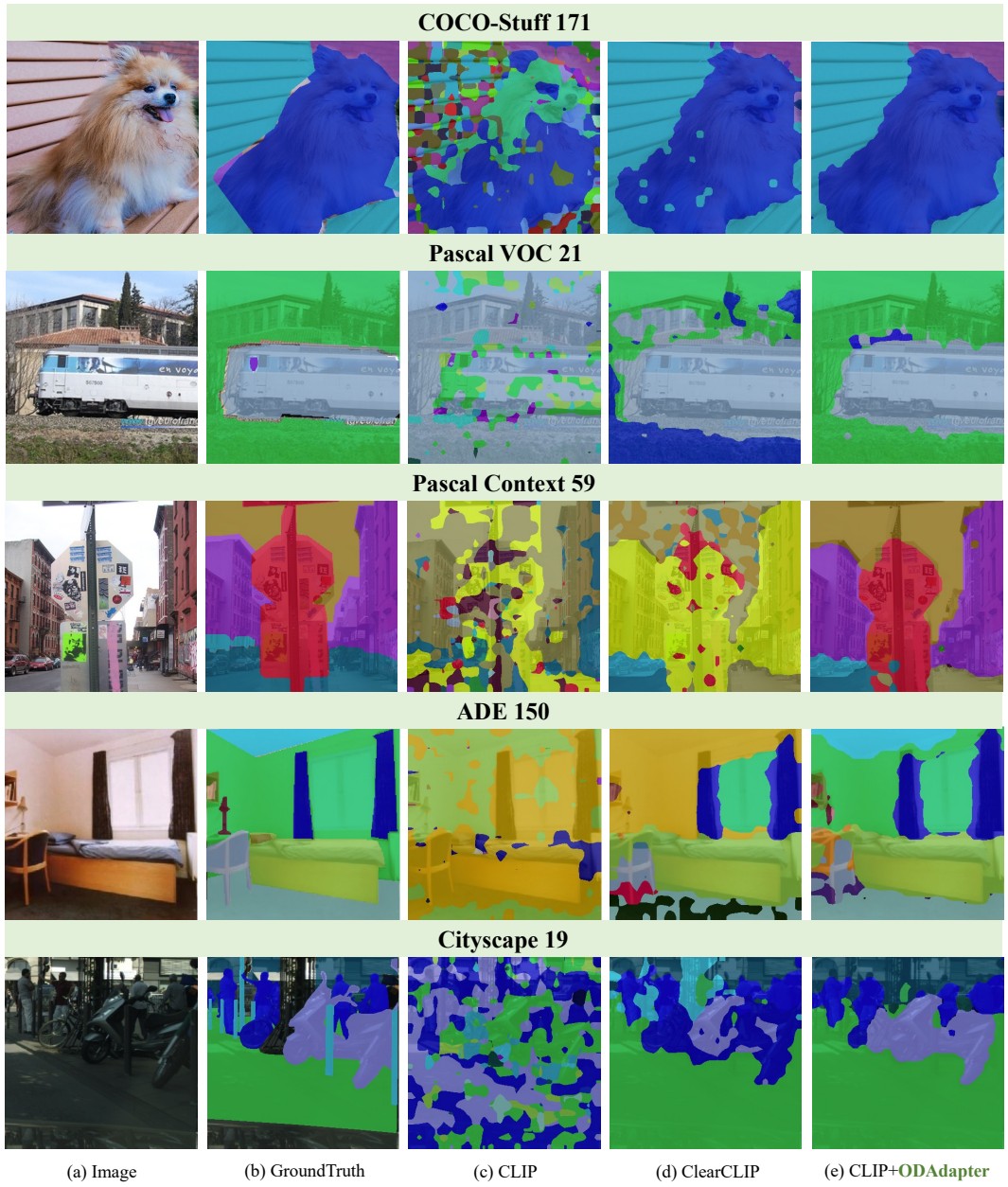

Figure 5: Visualization results on different datasets.

## A.2 Additional ablations

**Ablation on the choice of top k within SAA.** In our SAA module, the parameter $k$ of `topk` operation (Equation 11) controls how many of the most relevant text embeddings are allowed to interact with the visual features and subsequently influence the textual features through feedback. As shown in Table 6, when $k$ is set greater than or equal to the total number of categories, the mechanism degenerates into standard cross-attention, allowing all textual features to interact with the visual features. Conversely, when $k$ is set to the minimum value of 1, the interaction is limited to the single textual token corresponding to CLIP's global classification output. As shown in Table 6, setting $k$ too small (e.g., $k = 16$) or too large (e.g., $k = 128$) leads to performance degradation—likely due to the omission of relevant categories or the inclusion of redundant ones, respectively. These results highlight that a properly chosen $k$ value enhances the predicted attention map of CAI by improving pixel-level text-image alignment.

Table 6: Ablation on the choice of top k of SAA.

| Settings | ADE | City | Context59 | Object | VOC | Avg. |
|---|---|---|---|---|---|---|
| k=16 | 16.9 | 31.7 | 37.2 | 35.3 | 83.2 | 40.9 |
| k=32 | 17.6 | 32.4 | 38.2 | 36.2 | 83.6 | 41.6 |
| k=64 | 18.1 | 32.9 | 38.6 | 36.5 | 84.7 | 42.2 |
| k=96 | 17.7 | 32.5 | 38.4 | 36.3 | 84.1 | 41.8 |
| k=128 | 17.1 | 32.2 | 37.9 | 35.8 | 83.7 | 41.3 |

**Ablation on the components of SAA.** We also conduct an ablation study on the three key steps of the SAA module. For convenience, we refer to Text-Guided Visual Enhancement, Visual-Guided Text Refinement, and Cross-Modal Attention Refinement as TGV, VGT, and CMA, respectively. As shown in Table 7, both the text-to-image and image-to-text fusion steps contribute to improved alignment between modalities. This bidirectional interaction enables the injection of more accurate dense representations into CAI while preserving the original open-vocabulary capability of CLIP.

Table 7: Ablation on the components of SAA. TGV = Text-Guided Visual Enhancement, VGT = Visual-Guided Text Refinement, CMA = Cross-Modal Attention Refinement.

| Settings | ADE | City | Context59 | Object | VOC | Avg. |
|---|---|---|---|---|---|---|
| TGV | 17.5 | 32.2 | 37.8 | 36.0 | 83.3 | 41.4 |
| TGV+ CMA | 17.6 | 32.6 | 38.1 | 36.2 | 83.8 | 41.7 |
| TGV + VGT + CMA | 18.1 | 32.9 | 38.6 | 36.5 | 84.7 | 42.2 |

**Ablation on the choice of training dataset for OPMapper.** In addition to training OPMapper on COCO-Stuff, we also explore the use of other datasets commonly employed in dense prediction tasks. Specifically, we replace the original COCO-Stuff segmentation annotations with a mixture of annotations from Object365 Detection [41] and LVIS Segmentation [22]. This modification necessitates some adjustments to the training pipeline: 1). When constructing the prior attention maps for CAI, we treat all pixels within each bounding box in the detection datasets as belonging to a single class, and proceed to compute the prior attention maps in the same manner as with segmentation data. 2) During training, the SAA module is supervised using both segmentation and detection annotations. In contrast to supervision derived solely from segmentation, we additionally introduce a parallel detection head to predict bounding boxes. As shown in Table 8, even when using the mixture data, the model is still capable of effectively learning a mapper that transforms object-level query-key representations into pixel-level ones. These results suggest that training a standalone mapper appended to CLIP outputs is a viable strategy for enhancing its compatibility with open-vocabulary dense prediction tasks.

Table 8: Ablation on the choice of training dataset for OPMapper.

| Training Data | ADE | City | Context59 | Object | VOC | Avg. |
|---|---|---|---|---|---|---|
| COCO-Stuff | 18.1 | 32.9 | 38.6 | 36.5 | 84.7 | 42.2 |
| LVIS + O365 | 18.9 | 33.5 | 39.2 | 37.1 | 85.5 | 42.8 |

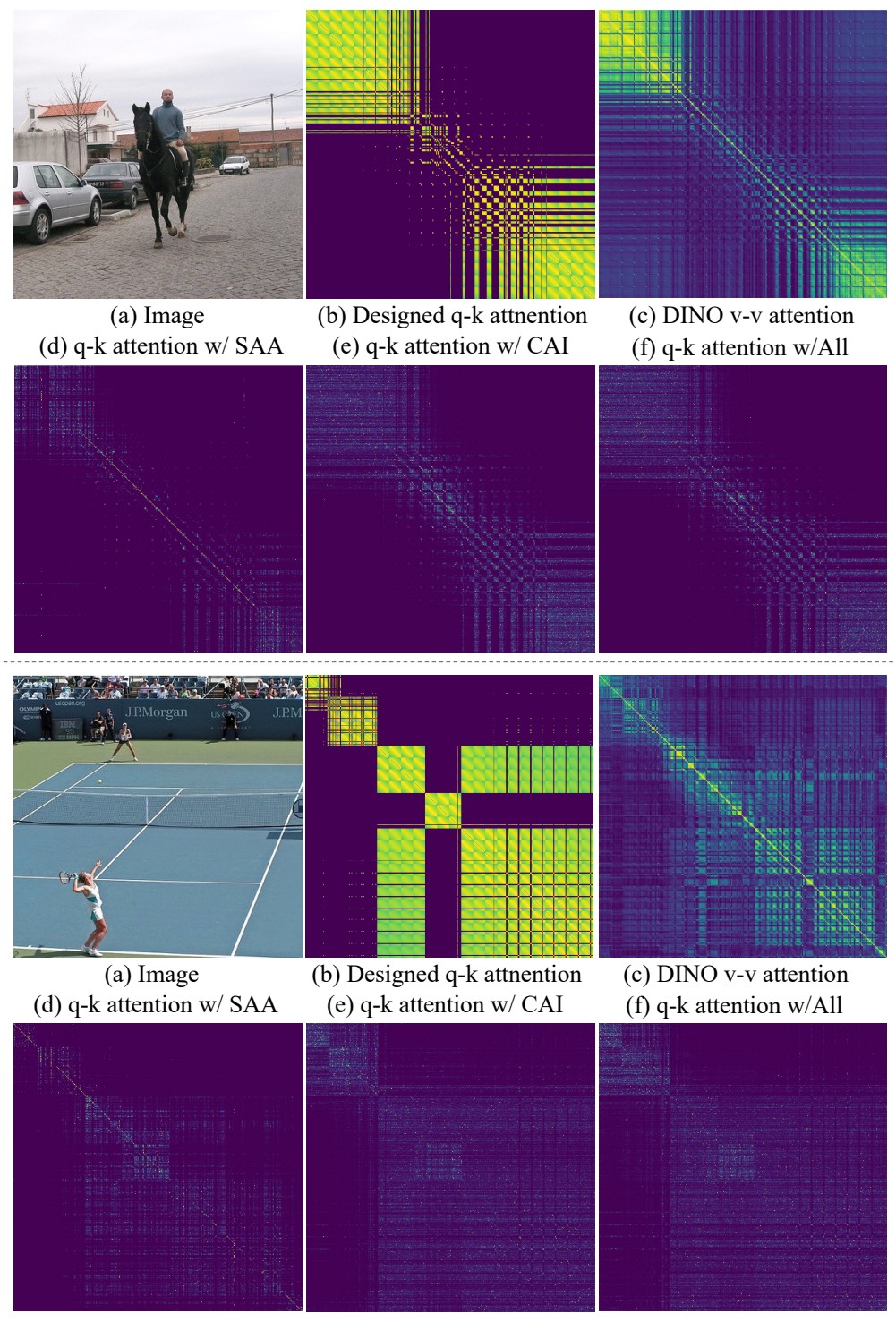

Figure 6: The visualizations of different types of $q$-$k$ attention weights. We applied normalization to the designed $q$-$k$ attention (Figure 6 b) and DINO $v$-$v$ attention weights (Figure 6 c), which enhances the brightness of the visualization and makes the patterns easier to observe (with the sum of each row exceeding 1). In contrast, the results produced by our OPMapper retain their original form, where the sum of each row equals 1. This design choice aims to clearly reflect the magnitude of the original values. For better clarity, it is recommended to view the corresponding figures in an enlarged format.

### A.3 Visualization Results of attention weights

In this section, we present visualizations of the predicted $q$-$k$ attention map from CAI, the corresponding designed attention map, and the prediction results, as illustrated in Figure 6. Specifically, we examine the following scenarios:

- Only CAI is applied (w/CAI).
- Only SAA is applied (w/SAA).
- Both CAI and SAA are applied (w/All), alongside a comparison with the $v$-$v$ attention map generated by DINO [4].

From these visualizations, it is obvious that the predicted attention weights under the supervision of CAI closely resemble the learned attention weights indicating their alignment. Our OPMapper produces attention maps that closely mirror those of the vision foundation model, such as DINO, further substantiating our approach and supporting our claims.

## B  Further analysis

### B.1  In-depth analysis of the motivation

The motivation of OPMapper: existing methods for computing attention weights mainly emphasize local compactness ($q$–$q$, $k$–$k$, or $q$–$q$ & $k$–$k$) while neglecting or down-weighting global connectivity. To overcome this, we propose the CAI module, which explicitly combines local compactness and global connectivity to produce pixel-aware attention maps. In parallel, the SAA module guides CAI in preserving CLIP's cross-modal alignment, thereby enabling open-vocabulary segmentation.

To validate the limitations of existing methods regarding global connectivity and to demonstrate that our approach better addresses this aspect, we conducted an experiment on 50 randomly selected images. Using the attention matrices from training-free methods such as MaskCLIP, we measured the attention between each token and same-category tokens at different spatial distances. Tokens were defined as "long-range" if their distance exceeded the average distance to all same-category tokens. For each image, we summed the attention values of these long-range tokens and then averaged across the image to obtain the mean long-range attention value. A higher value indicates greater emphasis on global connectivity.

Table 9: Long-range attention values for different models.

| Model | Mean long-range attention value |
|---|---|
| MaskCLIP [15] | 0.084 |
| SCLIP [44] | 0.142 |
| ClearCLIP [27] | 0.117 |
| ProxyCLIP [28] | 0.289 |
| CDAM [24] | 0.231 |
| OPMapper | 0.373 |

As shown in Table 9, our method achieves the highest mean long-range attention strength (0.373), while most other methods fall between 0.08 and 0.15, with the maximum only 0.289 (when incorporating external models such as DINO). These results indicate that our method captures more information from same-category but distant tokens, ensuring stronger global connectivity. This supports our argument that existing methods emphasize local compactness while underrepresenting global connectivity, whereas our CAI module explicitly strengthens global connectivity to construct more effective attention relationships. In addition, we visualize both our designed attention weights and the v–v attention produced by DINO in Figure 6. For clearer comparison, we applied brightness enhancement (so the sum of each row is not equal to 1). The visualization shows that in DINO's v–v attention, brighter values mainly appear near the diagonal of the matrix, indicating that local compactness dominates when merging information from other tokens, even though DINO's training emphasizes global connectivity more than other training-free methods.

## B.2 Deeper analysis of the prior attention weight

We clarify that our prior attention matrix design is not a simple intuition but fundamentally grounded on two well-established theoretical principles widely recognized in semantic segmentation literature:

**Local Compactness**: Fundamentally, semantic segmentation can be conceptualized as a problem of spatially continuous local clustering, wherein adjacent pixels are not independent but instead exhibit strong categorical coherence due to the inherent spatial smoothness of natural images. This property underscores the expectation that meaningful segmentation should leverage local contextual regularities, thereby reinforcing the robustness and consistency of the learned representations. This is commonly agreed in classical segmentation literature [37, 6]. In the main paper, Equation 4 explicitly incorporates the Euclidean distance into the attention computation to reinforce local semantic smoothness.

**Global Connectivity**: High-quality semantic segmentation fundamentally relies on the capacity to capture long-range semantic dependencies, beyond local contextual cues. This capability ensures that spatially distant pixels of the same semantic category—such as occluded regions of an identical object—are jointly recognized and consistently segmented [25]. Without modeling such non-local associations, segmentation results risk fragmenting semantically coherent regions, thereby undermining both accuracy and robustness. Vision transformers [36, 16] have also demonstrated this principle in modeling semantic relations between distant tokens and our Equation 5 precisely formulates a semantic-aware global connectivity pattern that aligns perfectly with this principle.

To further provide rigorous empirical evidence, we conducted the following experiment: we randomly selected 50 images containing open-set categories. For each image, we generated three attention maps using our proposed construction method, Cat-Seg (which was trained on other close-set categories but not trained on these open-set categories), and Mask2Former [8] (which was trained on both close-set and open-set categories, and thus its attention map is treated as the "optimal" one), respectivlly. We then computed both the average cosine similarity and average Structural Similarity Index (SSIM) to measure the similarity between two attention map. As shown in the Table 10, ours (prior attention matrix), average cosine similarity = 0.834 and average SSIM = 0.808, shows a consistently high similarity compared to Cat-Seg, rigorously demonstrating that our prior attention construction is much more similar to the fully-supervised attention structure.

Table 10: Comparison on cosine similarity and SSIM.

| Setting | Cosine Similarity | SSIM |
|---|---|---|
| Ours vs Mask2Former | 0.834 | 0.808 |
| Cat-Seg vs Mask2Former | 0.785 | 0.771 |

## B.3 Discussion on the difference with related works

In general, the originality of OPMapper and its distinctions from other related methods are reflected in the following aspects:

- We propose an explicitly designed local–global composite attention weight construction that directly maps $q$–$k$ to obtain pixel-aware attention weights. In contrast, other methods compute attention either by incorporating external modules or by relying on self–self attention that primarily enhances local compactness. These two categories of approaches only achieve pixel-aware features indirectly or approximately.

- We introduce the cross-modal semantic alignment module (SAA), which leverages alternating text–image fusion as an implicit supervision signal to guide the learning of the mapping module (i.e., CAI), while preserving CLIP's original vision–language alignment. Other methods, however, neglect maintaining this alignment and disregard the utilization of the CLIP text encoder.

- Our method is modular, lightweight, and incurs no additional overhead during inference, making it adaptable to a wide range of approaches. In contrast, other methods lack such flexibility and generalizability, limiting their applicability across diverse settings.

One similar solution as a plug-and-play modular for other open-vocabulary semantic segmentation method is CDAM [24]. It leverages the category distribution correlations in the image–text similarity map to construct attention better suited for precise localization, and it achieves strong results in experiments. However, our method and CDAM differ significantly in terms of motivation and innovations, model framework, adaptation process, and experimental characteristics. In the tableX, we summarize the approaches of both methods across these dimensions and highlight the key differences.

Table 11: Comparison between OPMapper and CDAM.

| Dimension | OPMapper | CDAM |
|---|---|---|
| Motivation & innovations | We observe that frozen CLIP features are inherently object-aware. Under consistent value representations, **transforming the attention into pixel-aware form facilitates direct transfer to dense tasks**. The learning of the mapper, however, must **balance local compactness and global connectivity, while preserving CLIP's inherent vision–language alignment.** | They observe that for each token (corresponding to a patch), **its classification vector distribution exhibits category dependency. Leveraging this distributional property allows them to compute more robust inter-token correlations**, ultimately yielding an attention mechanism better suited for segmentation tasks. |
| Architecture | A Plugin Mechanism. CAI is trained with manually designed ground-truth attention and guided by the SAA module to learn how to map $q$ and $k$. **In inference, the $q$ and $k$ from the final CLIP layer are mapped into $q'$ and $k'$, which yield pixel-aware attention.** Together with the frozen $v$, these form the image features, which are then integrated with text features to produce the final mask. | A Rewriting Mechanism. **Using the frozen $v$ from the final CLIP layer and the enhanced text, CDAM derives a category distribution vector for each token. Attention is then computed by measuring the differences among these token-level distributions.** With this attention, $v$ is recalculated to obtain a new $v'$, which is subsequently combined with the original (non-enhanced) text features to generate the mask. |
| Image-Text Alignment | SAA performs iterative alignment of image and text features, implicitly guiding OPMapper and ultimately influencing $q$ and $k$. | CDAM lacks both explicit and implicit vision–language alignment mechanisms. As a result, the attention generated from category distribution similarity, when used together with the value features to compute $v'$, may not be compatible with the frozen text features. |
| Adaptation process | Adapt OPMapper to $q$ and $k$ of the final CLIP layer to obtain new $q'$ and $k'$, and then calculate image features through conventional dot-product attention with frozen $v$. | Compute similarity between text and image features, and then calculate attention by applying JS divergence to the similarity map. Finally, use the attention map and frozen $v$ to calculate the final image features. |
| Threshold sensitivity in complex scenes | OPMapper requires **no threshold** or heuristic hyperparameter and learns attention weights in a fully end-to-end manner. | CDAM relies on a **manually selected entropy-based threshold** to filter category distributions. As acknowledged by the authors, this threshold is difficult to determine in complex scenes. |
| Differences in global semantic modeling | OPMapper, by design, **assigns high attention weights explicitly to distant same-category tokens, ensuring more consistent global connectivity.** | CDAM enhances correlations among nearby same-category patches using JS divergence, but **its handling of long-range same-category tokens still depends on thresholding.** |
| Applicability scope | OPMapper is compatible with both training-free and training-based segmentation frameworks, showing broader versatility. | CDAM is currently only applicable to training-free methods. |

# C  Details of Mapper

We provide a detailed description of the Mapper's architecture in Figure 7. As shown in Figure 2(b), the CAI consists of two parallel components, Mapper A and Mapper B, which share the same structure but operate with independent parameters. Taking Mapper A as an example, it incorporates two modified ResAttn blocks compared to the original CLIP [40] architecture. These modifications are designed to enhance the mapper's functionality while maintaining computational efficiency.

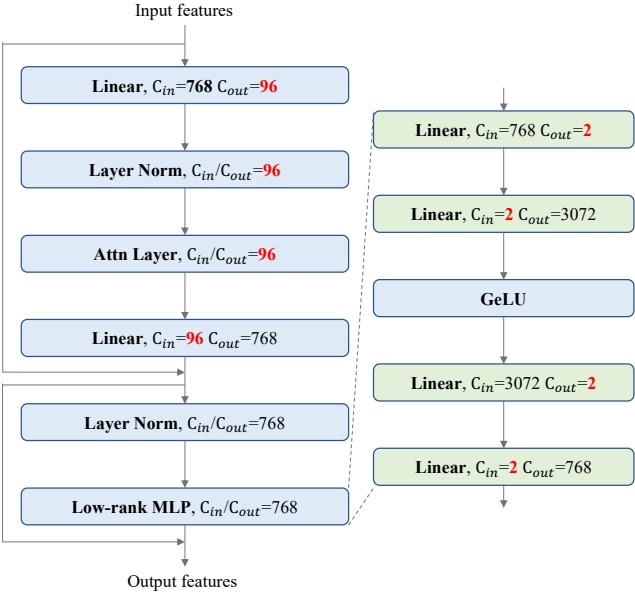

Figure 7: The detailed architecture of the modified ResAttnBlock used in our CAI is illustrated. The components highlighted in red indicate the differences compared to the original ResAttnBlock. Additionally, the green Linear represents a low-rank linear transformation with a rank of $r = 2$.

