# OpenReview forum: "OPMapper: Enhancing Open-Vocabulary Semantic Segmentation with Multi-Guidance Information"
_NeurIPS.cc/2025/Conference — NeurIPS 2025 poster_

### Official Review · Reviewer_WxDe · 2025-06-13

**Clarity:** 2
**Significance:** 2
**Originality:** 2
**Rating:** 4
**Confidence:** 5

**Summary:**

This paper proposes a plug-and-play module, which is trained with the help of pixel-level annotated data to improve the dense representation capability of CLIP. This paper argues that training-based methods consume large amounts of resources, and training-free methods favor local smoothness while overlooking global semantics. The key contribution is to design self-attention labels to supervise the last layer of CLIP’s self-attention matrix to focus on local-global relationships.

**Questions:**

please refer to weakness.

**Ethical Concerns:**

["NO or VERY MINOR ethics concerns only"]

**Final Justification:**

The author has provided a very detailed and reliable analysis of the issues that concern me. I decided to raise my score to 4.

**Limitations:**

yes

**Quality:**

2

**Strengths And Weaknesses:**

Strengths:

1. The methods section is clearly written.
2. Fig.2 is clear.

Weaknesses:

1.	The proposed method seems not to address the motivation that training-based method demands large-scale pixel-level annotations and significant GPU hours (Line.31). KL loss between manually designed attention and CLIP’s attention seems to require more GPU memory than the regular pixel-wise cross entropy loss (Because the self-attention matrix is very large). Please offer more training cost analysis.
2.	The claim that training-free method favor local smoothness while overlooking global semantics (Line.40-45,5-6) needs to be more explored. Please provide some quantitative or qualitative insights to support the rationale for the motivation of the proposed method.
3.	Why not calculate KL loss on the normalized distribution? Is Eq.6 softmax normalized? In addition, $W_p=Q^\prime K^{\prime \top}$ seems not to softmax normalized, is the training stable if KL loss is calculated directly without normalization?
4.	More experiment exploration about $\lambda$. Such as setting it to learnable scalar. For high frequency areas like boundaries, does $\lambda$ need to be larger? Please provide more insights on the local-global tradeoff (as this is one of the motivations).
5.	The proposed method seems to aim to train CAI and discard SAA when using, so why not just use a linear layer instead of SAA? Please provide some insights. If the proposed method hopes to provide a trained segmentation model in addition to training a plug-in CAI, then the experimental results of the whole framework should be given to compare with the existing training-based methods.
6.	Scaling experiments about training-based method needs to be provided.

---

> ### Author Rebuttal · Authors · 2025-07-31
>
> Dear reviewer WxDe,
>
> We sincerely thank you for your thoughtful comments, valuable insights, and the considerable time and effort you have dedicated to carefully reviewing our manuscript. The concerns and suggestions raised by you have provided us with an excellent opportunity to significantly enhance the quality and clarity of our paper. In the following, we address each concern point-by-point and present additional analyses and experiments to thoroughly resolve your concerns.
>
> &nbsp;
>
> ---
>
> ### 1. W1: More training cost analysis
>
> We thank the reviewer for raising concerns regarding training efficiency. It is worth noting that training-based approaches typically require updating the backbone or relatively heavy fusion/segmentation modules, which constitute the most time- and memory-intensive components. In contrast, our method, when training OPMapper, only requires training two lightweight modules, CAI and SAA.
>
> **Both CAI and SAA are designed to be extremely efficient**, such that the entire training can be completed with less than 2GB of GPU memory while processing 8 images per GPU. Therefore, **the difference in GPU memory consumption between using KL divergence and pixel-wise cross-entropy is practically negligible in our current setting**. Moreover, pixel-aware cross-entropy loss is also not well-suited for measuring the discrepancy between two distributions.
>
> Under current settings, the training takes approximately 3 hours with ViT-B and 4 hours with ViT-L. Furthermore, once trained, these modules can be seamlessly integrated into other models without additional retraining.
>
> To provide a more comprehensive comparison, we present the training costs of representative training-based methods in the table below. The results clearly indicate that, while enhancing generalizability, our method achieves a substantial reduction in training costs, including both training time and GPU memory consumption.
>
> |model|backbone|Batch size x GPUs|Training time|VRAM|iterations|
> |-----|--------|-----------------|-------------|----|----------|
> |Cat-seg|ViT-B|4*4| ~8h | ~23GB | 80k |
> |SCAN | ViT-L |4*8 | ~18h | ~30GB | 80k |
> |SAN | ViT-B | 4*8 | ~18h | 6.6GB | 60k |
> |Ours | ViT-B | 8*4 | ~3h  | <2GB  | 80k |
> |Ours | ViT-L | 8*4 | ~4h  | <4GB | 80k |
>
> &nbsp;
>
> ---
>
> ### 2. W2: In-Depth Analysis of the Motivation
>
> We first clarify the motivation of this work: existing methods, when computing attention weights, primarily emphasize local compactness (q-q, k-k or q-q&k-k attention) while neglecting or down-weighting global connectivity. To address this limitation, we introduce the CAI module, which explicitly integrates both local compactness and global connectivity to directly output pixel-aware attention maps. In parallel, we employ the SAA module to guide CAI in preserving CLIP’s inherent cross-modal alignment, thereby enabling open-vocabulary segmentation.
>
> To validate the drawbacks of other methods in regard to global connectivity and demonsrate that our method better accounts for global connectivity compared to existing approaches, we conducted the following experiment on 50 randomly selected images. Specifically, using the attention matrices from training-free methods such as MaskCLIP and ProxyCLIP, we measured the attention value between each token and same-category tokens at varying spatial distances. We defined tokens as “long-range” if their spatial distance exceeded the average distance from the given token to all same-category tokens. For each image, we aggregated the attention values of these long-range tokens and computed their total value, followed by averaging across the entire image to obtain the mean long-range attention value. A higher value indicates greater emphasis on global connectivity. The results are presented in the table below.
>
> | Model | mean long-range attention value |
> |-|-|
> |MaskCLIP|         0.084     |
> |SCLIP |    0.142            |
> |ClearCLIP|     0.117              |
> |ProxyCLIP|      0.289             |
> |Ours |        0.373               |
>
> As shown in the table, our method achieves the highest mean long-range attention strength, reaching 0.373, while most other methods fall within the range of 0.08 to 0.15, with the maximum being only 0.289 (achieved when incorporating external models such as DINO to provide global connectivity). These results demonstrate that our method effectively captures more information from same-category but spatially distant tokens, thereby ensuring more comprehensive global connectivity. This further substantiates our argument that existing methods tend to emphasize local compactness while underrepresenting global connectivity, whereas our CAI module explicitly enhances global connectivity, thereby achieving a more effective construction of attention relationships.
>
> In addition, we kindly refer the reviewer to Figure 2 in our supplementary material. There, we visualize both our designed attention weights and the v-v attention produced by DINO. To facilitate a clearer comparison, we applied brightness enhancement (so that the sum of each row is not equal to 1). We also sincerely apologize for the typo appearing in the Figure 2 and will correct this in our paper. From the visualization, it is observed that in DINO’s v-v attention, the brighter values predominantly appear near the diagonal of the attention matrix, indicating that local compactness is the dominant factor when merging information from other tokens, even though global connectivity is heavily emphasized during DINO’s training compared with other training-free methods.
>
> &nbsp;
>
> ---
>
> ### 3. W3: Distribution of $W_{qk}$ and $W_{p}$ in Eq 6
>
> Thank you for highlighting this important point. We agree that clarification regarding the normalization of Eq. (6) is necessary. Specifically, the components $W_l$ and $W_g$ in Eq. (6) are indeed normalized distributions: $W_l$ is normalized via a standard softmax operation (line 173), ensuring each row sums to 1. Similarly, $W_g$ is constructed as a uniform distribution (line 179) among pixels of the same category, which can also be viewed as a special case of softmax normalization, guaranteeing each row sums exactly to 1 as well.
>
> Since the weights in Eq. (6) (i.e., $\lambda$ and $1-\lambda$) sum to 1, the resulting combined attention matrix $W_{qk}$ is inherently normalized, ensuring each row also sums to 1. Regarding the predicted attention $W_p$ , it is also normalized via a softmax operation during training. However, we acknowledge that the explicit softmax symbol was omitted in the manuscript, potentially causing confusion. We will clearly include this explicit normalization step in future manuscript revisions to avoid any misunderstanding.
>
> &nbsp;
>
> ---
>
> ### 4. W4: The choice of $\lambda$
>
> We have conducted additional experiments on the choice of $\lambda$ in Table 2 of the supplementary material to further validate the rationale behind our selection. We kindly invite the reviewer to refer to these results for more details. Moreover, following the reviewer’s suggestion, we also performed a quick experiment by treating $\lambda$ as a learnable scalar. The result shows that when $\lambda = 0.3372$, the average performance reaches 41.9, which is very close to the manually set value of 0.3. This further substantiates the effectiveness of our ablation study.
>
> Regarding high-frequency regions (e.g., boundaries), since they violate the assumption of local smoothness, the corresponding local weight $\lambda$ is set to be **smaller** in these areas.
>
> &nbsp;
>
> ---
>
> ### 5. W5: The design of SAA
>
> First, we would like to clarify that our method does NOT rely on providing a trained segmentation module to achieve the final segmentation. The ONLY component inserted into other methods is the CAI module.
>
> Second, we address the question of why we designed the SAA module instead of simply using an MLP. During the training of CAI, the most direct supervision signal comes from our manually constructed “ideal” attention matrix, i.e., the q-k weight map, which serves as an explicit form of supervision. The purpose of SAA, however, is to provide an indirect, implicit supervisory signal. Specifically, the predicted q-k weights are combined with the original v from the frozen CLIP to form the input to SAA, which then generates a segmentation map for loss computation. In the process of transforming input into output, SAA iteratively performs image-to-text and text-to-image fusion, ensuring proper alignment of the two modalities in CLIP. This alignment is propagated back to CAI through the predicted q-k weights. Therefore, SAA functions more like a mechanism for computing the loss, enabling CAI outputs to not only achieve the transition from object-level to pixel-level representations but also preserve the original cross-modal alignment. A simple MLP cannot fulfill this design objective.
>
> To further verify this, we conducted an experiment where we replaced SAA with MLPs of approximately the same parameter size, applying it to transform the q-k weights, while keeping the subsequent segmentation generation process identical. When using ViT-B, this model achieved an average mAP of 40.5 across five datasets (ADE, Cityscapes, Context59, Object, and VOC-20). In comparison, the CAI+SAA combination achieved an average mAP of 42.2, while CAI without SAA yielded 41.3. These results demonstrate that directly using an MLP is suboptimal, as it even suppresses part of the improvement brought by CAI itself (41.3 → 40.5).
>
> &nbsp;
>
> ---
>
> ### 6. W6: Scaling experiments
>
> Following reviewer's advice, we conducted a model scaling experiment by inserting the ViT-L version of CAI into Cat-Seg. As shown, this integration led to improvements of +0.6, +1.5, +0.4, and +0.5 mIoU on the VOC, Context-59, Stuff-171, and ADE-150 benchmarks, respectively.
>
> &nbsp;
>
> We hope that our response has satisfactorily addressed your concerns, and we thank you again for your thoughtful feedback.
>
> &nbsp;
>
> Best regards,
>
> Authors

---

> > ### Comment · Reviewer_WxDe · 2025-08-02
> >
> > Thank you for your patient reply, most of my questions have been solved. However, there is still a key issue I would like to discuss with you: the loss supervision on self-attention matrices.
> >
> > 1.Assuming the space complexity of traditional cross-entropy loss is O(N) (where N denotes the total number of pixels), should the KL divergence loss between self-attention matrices be characterized as O(N²)? If so, how does this align with your reported GPU memory consumption of under 2GB in Table 3? While the module is lightweight, the loss function's memory footprint warrants examination, especially considering techniques like Mask2Former's point sampling that reduce memory by computing loss on subsets rather than all pixels.
> >
> > 2.The label design of the self-attention matrix seems a bit confusing, and the design method is somewhat in line with intuitive cognition, and does not have a good inspiration.Are there any interpretable experiments to explain the rationale for this label design?

---

> > > ### Author Response · Authors · 2025-08-02
> > >
> > > Dear reviewer,
> > >
> > > Thank you for your feedback. We kindly ask for your patience, and we will provide you with a more detailed response shortly.
> > >
> > > Best regards,
> > >
> > > Authors

---

> > ### Comment · Reviewer_WxDe · 2025-08-06
> >
> > Thanks to the author for his serious reply. My question has been resolved and I decided to raise the score.

---

> > > ### Author Response · Authors · 2025-08-06
> > >
> > > Dear reviewer WxDe,
> > >
> > > We sincerely appreciate your valuable reviews, your  recognition of our response and your positive evaluation of our work. Wishing you joy and pleasant moments every day.
> > >
> > > Best regards,
> > >
> > > Authors

---

> ### Author Response · Authors · 2025-08-03
> **Discussion - (1/2)**
>
> Dear reviewer WxDe:
>
> We sincerely apologize for the delay, and we are very glad to discuss with you the details of attention matrix training. Below, we provide a point-by-point clarification.
>
> ### 1. Memory cost of KL loss.
>
> The reason our method has a relatively small memory usage can be explained as follows:
>
> 1. The parameters of ViT-B are completely frozen and do not participate in any training.
> 2. There might be a slight misunderstanding: **our KL divergence is not computed at the pixel level, but at the token level, since the attention map is token-based rather than pixel-based**. Thus, the $n$ of O($n^2$) is 576.
> 3. Because each row of the attention map represents a complete distribution and is independent from the others, we are able to perform row-wise streaming computation.
> 4. Our CAI module is extremely lightweight, containing only 0.8M parameters.
>
> Next, we provide a detailed example for calculation. Based on the ViT-B/16 model with an input resolution of $384 \times 384$, the final layer outputs $24 \times 24 + 1 = 577$ tokens. Since we exclude the CLS token when computing the attention map, the sizes of q and k are both 576, resulting in an attention map of size $576 \times 576$. The batch size equals 8 and we use fp32. Based on these values, the memory consumption during training is approximately:
>
>
> | Module                           | Weights (MB) | Gradients (MB) | Adam (MB) | Activations (MB) | Total (MB) |
> |----------------------------------|--------------|----------------|-----------|------------------|------------|
> | Frozen ViT-B/16                  | 86 M params × 4 B = 344          | 0              | 0         | 0                | **344**    |
> | CAI (2 mappers × 2 blocks)       | 0.83 M params × 4 B = 3.3          | 3.3            | 0.83 M params × 4 B x 2  = 6.6       | 6.75 MB/mapper × 2 x 8(bs) = 54                | **67**     |
> | SAA (3 cross-attn blocks)        | 7.09 M params × 4 B x 3 = 85.2         | 85.2           | 7.09 M × 4 B x 2 x 3 = 170.4     | 15 MB/block (with checkpoint) × 3 x 8(bs) = 360              | **701**    |
> | Losses (KL + DICE + BCE)         | 0            | 0              | 0         | KL: $576^2$ * 2 * 4 B * 8(bs) ≈ 20.8  (all preserved, not row-streamed, target **detached**) + (DICE + BCE): $96²$ × 8 × 4 B -> 50.6 (safe upper bound) = 71                | **71**     |
> | **Subtotal**                     | —            | —              | —         | —                | **1 183**  |
> | **+ 15 % buffer / fragmentation**| 1183 MB × 1.15            |               |          |                 | **1 360**  |
> | **Peak per GPU**                 | —            | —              | —         | —                | **≈ 1.36 GB** |
>
> **If we switch to cross-entropy (CE) loss, the memory usage would be the same as the row-streaming implementation of KL loss (which means we can achieve memory usage comparable to CE loss), i.e., $20.8 / 576 = 0.0361$ MB. Within the overall memory consumption, KL loss accounts for only $20.8 / 1360 = 0.015$. Under this setting, the additional memory overhead introduced by KL loss compared to CE loss is negligible.**
>
> Therefore, although the complexity of KL divergence is indeed $O(n^2)$ as you pointed out, the value of $n$ remains sufficiently small (576), and the overall network is extremely lightweight, enabling us to train CAI with only 2GB of GPU memory. We kindly invite the reviewer to consult Figure 1 in our supplementary material for a clearer understanding of the CAI module.
>
> &nbsp;
>
> **Due to space limitations, we kindly refer the reviewer to another response for the second point.**

---

> ### Author Response · Authors · 2025-08-03
> **Discussion - (2/2)**
>
> ### 2. The design and interpretability of the ideal attention map
>
> Your question is indeed very meaningful. However, we would like to clarify that our attention map is not a self–self attention. Instead, it is computed from the query and key after transformation by CAI.
>
> Next, I will address the inspiration behind our design. This point was also discussed in our Response 1 to Reviewer ohL3, and for completeness, I will reproduce that explanation here.
>
> Our design is not a simple intuition but  fundamentally grounded on two well-established theoretical principles widely recognized in semantic segmentation literature:
>
> 1. Local Compactness: Fundamentally, semantic segmentation can be conceptualized as a problem of spatially continuous local clustering, wherein adjacent pixels are not independent but instead exhibit strong categorical coherence due to the inherent spatial smoothness of natural images. This property underscores the expectation that meaningful segmentation should leverage local contextual regularities, thereby reinforcing the robustness and consistency of the learned representations. This is commonly agreed in classical segmentation literature (FCN [Long et al., CVPR’15], DeepLab [Chen et al., TPAMI’18]). In our paper, Equation 4 explicitly incorporates Euclidean distance into attention computation to reinforce local semantic smoothness.
>
> 2. Global Connectivity: High-quality semantic segmentation fundamentally relies on the capacity to capture long-range semantic dependencies, beyond local contextual cues. This capability ensures that spatially distant pixels of the same semantic category—such as occluded regions of an identical object—are jointly recognized and consistently segmented (AAF [Ke et al., ECCV'18]). Without modeling such non-local associations, segmentation results risk fragmenting semantically coherent regions, thereby undermining both accuracy and robustness. Vision transformers (ViT, Swin Transformer) have also demonstrated this principle in modeling semantic relations between distant tokens and our Equation (5) precisely formulates a semantic-aware global connectivity pattern that aligns perfectly with this principle.
>
> To further address the reviewer’s concern with rigorous empirical evidence, We conducted the following experiment: we randomly selected 50 images containing open-set categories. For each image, we generated three attention maps using our proposed construction method, Cat-Seg (which was trained on other close-set categories but not trained on these open-set categories), and Mask2Former (which was trained on both close-set and open-set categories, and thus its attention map is treated as the "optimal" one), respectivlly. We then computed both the average cosine similarity and average Structural Similarity Index (SSIM) to measure the similarity between two attention map. As shown in the table:
>
> | Setting  |  Cosine Similarity  |  SSIM  |
> | --------- | -------------------- | ---------|
> |Ours vs Mask2Former   |     0.834      |          0.808 |
> |Cat-Seg vs Mask2Former  |     0.785    |     0.771 |
>
> **Ours (prior attention matrix), average cosine similarity = 0.834 and average SSIM = 0.808, shows a consistently high similarity compared to Cat-Seg, rigorously demonstrating that our prior attention construction is much more similar to the fully-supervised attention structure.**
>
> Furthermore, as clearly shown in Supplementary Figure 2 and the Figure 3 of our paper, the attention patterns generated by CAI supervision remarkably resemble those produced by vision foundation models (e.g., DINO), providing strong qualitative validation, and the segmentation results remain sharp along boundaries and coherent across object parts compared with other methods. These analyses collectively provide rigorous justification beyond mere intuition.
>
> &nbsp;
>
> We sincerely hope that our response will gain your recognition and foster further constructive feedback.
>
> &nbsp;
>
> Best regards,
>
> Authors

---

### Official Review · Reviewer_ReDz · 2025-06-24

**Clarity:** 2
**Significance:** 2
**Originality:** 2
**Rating:** 4
**Confidence:** 5

**Summary:**

This paper proposes OPMapper, a lightweight, plug-and-play module designed to enhance open-vocabulary semantic segmentation by improving the pixel-level attention maps of frozen CLIP models. OPMapper achieves this by injecting both local compactness and global connectivity into attention weights via Context-aware Attention Injection (CAI) and aligning them with textual prompts using Semantic Attention Alignment (SAA) during training. Integrated seamlessly into existing methods, OPMapper increases performance across various benchmarks with minimal computational overhead at inference time.

**Questions:**

1. A detailed analysis is needed to determine whether previous methods indeed suffer from weak global connectivity.

2. Please analyze whether OPMapper effectively addresses this global connectivity issue.

3. Despite incorporating training, the performance improvement on training-free methods is limited. Could the authors explain why?

4. Table 1 lacks comparison with more recent methods. Please justify the selection of baseline methods.

5. Why is KL divergence used in Equation (8)?

6. Is it valid to use W_qk as the reference target for OPMapper? Please provide justification.

7. Would it not be better for OPMapper to update both q and k, similar to conventional self-self attention, to create q–q and k–k attention?

8. In Equation (10), shouldn't the dimensions be R^C and R^(HW x C), instead of R^L and R^(HW x L)?

**Ethical Concerns:**

["NO or VERY MINOR ethics concerns only"]

**Final Justification:**

All of my concerns have been resolved and I change the score from 3 to 4.

**Limitations:**

Yes. They discussed the limitations and broader impact.

**Paper Formatting Concerns:**

In Line 147, I believe there should be additional vertical spacing above it to separate it more clearly from Line 146.

**Quality:**

3

**Strengths And Weaknesses:**

Strengths

1.OPMapper improves segmentation performance through various methods.

2.OPMapper is compatible with both training-based and training-free approaches, enhancing its versatility.

Weakness

1. Despite incorporating training part, the performance gains for training-free methods remain relatively limited.

2. The paper lacks a clear analysis of the limitations of existing methods and does not sufficiently justify the proposed approach. For instance, issues such as the lack of global connectivity in previous methods are not clearly examined.

3. Due to weak analysis and motivation, it is difficult to assess the significance of the paper’s main claims.

4. Existing methods also aim to improve attention maps, so it is unclear how OPMapper distinguishes itself in terms of originality.

---

> ### Author Rebuttal · Authors · 2025-07-31
>
> Dear reviewer ReDz,
>
> We sincerely thank you for your thoughtful comments, valuable insights, and the considerable time and effort you have dedicated to carefully reviewing our manuscript. In the following, We have summarized the related concerns into one unified concern, and address each unified concern point-by-point.
>
> &nbsp;
>
> ---
>
> ### 1. W1+Q3: limited performance improvement on training-free methods
>
> First, we would like to clarify that our CAI module is extremely lightweight, and importantly, its training process and loss functions are entirely independent of the segmentation modules and categories, respectively.
>
> Then, in our experiments, we observe clear improvements in average performance among 8 benchmarks across most models. The only exception is ProxyCLIP, where the improvement is about 1%. This is because ProxyCLIP already leverages more powerful external models (e.g., DINO, MAE) to optimize its attention weights. The parameter scale and training cost of these models are substantially larger than those of our CAI. Therefore, the relatively limited improvement (~1%) observed on ProxyCLIP is reasonable and consistent with expectations.  Importantly, **excluding ProxyCLIP, our method achieves gains ranging from 2% to 18.9% on other methods among 8 benchmarks, highlighting both the stability and the broad applicability of our approach**.
>
> &nbsp;
>
> ---
>
> ### 2. W2+W3+Q1+Q2: In-Depth Analysis of the Motivation: Why Prior Methods Lack It and How Our Approach Validates It
>
> We first clarify the motivation of this work: existing methods for computing attention weights mainly emphasize local compactness (q–q, k–k, or q–q & k–k) while neglecting or down-weighting global connectivity. To overcome this, we propose the CAI module, which explicitly combines local compactness and global connectivity to produce pixel-aware attention maps. In parallel, the SAA module guides CAI in preserving CLIP’s cross-modal alignment, thereby enabling open-vocabulary segmentation.
>
> To validate the limitations of existing methods regarding global connectivity and to demonstrate that our approach better addresses this aspect, we conducted an experiment on 50 randomly selected images. Using the attention matrices from training-free methods such as MaskCLIP, we measured the attention between each token and same-category tokens at different spatial distances. Tokens were defined as “long-range” if their distance exceeded the average distance to all same-category tokens. For each image, we summed the attention values of these long-range tokens and then averaged across the image to obtain the mean long-range attention value. A higher value indicates greater emphasis on global connectivity. The results are shown in the table below.
>
> |Model|mean long-range attention value|
> |-|-|
> |MaskCLIP|0.084|
> |SCLIP|0.142|
> |ClearCLIP|0.117|
> |ProxyCLIP|0.289|
> |Ours | 0.373|
>
> As shown in the table, our method achieves the highest mean long-range attention strength (0.373), while most other methods fall between 0.08 and 0.15, with the maximum only 0.289 (when incorporating external models such as DINO). These results indicate that our method captures more information from same-category but distant tokens, ensuring stronger global connectivity. This supports our argument that existing methods emphasize local compactness while underrepresenting global connectivity, whereas our CAI module explicitly strengthens global connectivity to construct more effective attention relationships.
>
> In addition, we kindly refer the reviewer to Figure 2 in our supplementary material, where we visualize both our designed attention weights and the v–v attention produced by DINO. For clearer comparison, we applied brightness enhancement (so the sum of each row is not equal to 1). We sincerely apologize for the typo in Figure 2 and will correct it in the paper. The visualization shows that in DINO’s v–v attention, brighter values mainly appear near the diagonal of the matrix, indicating that local compactness dominates when merging information from other tokens, even though DINO’s training emphasizes global connectivity more than other training-free methods.
>
> &nbsp;
>
> ---
>
> ### 3. W4: Difference with other methods that aim to improve attention maps
>
> We thank the reviewer for raising this concern. The originality of OPMapper and its distinctions from other methods are reflected in the following aspects:
>
> 1. We propose an explicitly designed local–global composite attention weight construction that directly maps q–k to obtain pixel-aware attention weights. In contrast, other methods compute attention either by incorporating external modules or by relying on self–self attention that primarily enhances local compactness. These two categories of approaches only achieve pixel-aware features indirectly or approximately.
>
> 2. We introduce the cross-modal semantic alignment module (SAA), which leverages alternating text–image fusion as an implicit supervision signal to guide the learning of the mapping module (i.e., CAI), while preserving CLIP’s original vision–language alignment. Other methods, however, neglect maintaining this alignment and disregard the utilization of the CLIP text encoder.
>
> 3. Our method is modular, lightweight, and incurs no additional overhead during inference, making it adaptable to a wide range of approaches. In contrast, other methods lack such flexibility and generalizability, limiting their applicability across diverse settings.
>
> &nbsp;
>
> ---
>
> ### 4. Q4: Baseline and comparison with recent methods
>
> We thank the reviewer for the valuable suggestion. The baselines we selected are widely cited, come with publicly available code, and are commonly used as SOTA benchmarks (e.g., ClearCLIP). We also chose CLIP as it is the foundational training-free model on which most subsequent methods build.
>
> Beyond training-free methods, we integrated our approach into training-based SOTA models as well, where we similarly observed consistent performance improvements.
>
> With respect to more recent methods, we identified two training-free approaches (LPOSS [CVPR’25], CASS [CVPR’25]) from CVPR 2025. Incorporating OPMapper into these methods, we achieved performance improvements, and we will include the corresponding experimental results in the revised manuscript.
>
> |model|VOC21|PC60|Object|VOC20|PC59|Stuff|City|ADE|
> |-|-|-|-|-|-|-|-|-|
> |LPOSS|60.2|35.0|34.7|80.2|36.9|25.3|37.6|21.2|
> |+OPMapper|+3.5|+0.7|+0.5|+4.7|+1.0|+1.1|+2.4|+0.9|
> |CASS|64.3|36.9|38.1|88.3|39.6|26.2|39.8|20.1|
> |+OPMapper|+2.7|+1.3|+0.7|+1.7|+0.5|+0.9|+1.2|+0.7|
>
> &nbsp;
>
> ---
>
> ### 5. Q5+Q6: The choice of using KL loss and the effectiveness of $W_{qk}$
>
> KL divergence is inherently well-suited for measuring the difference between two probability distributions, which makes it particularly appropriate for normalized probability-based representations such as attention matrices. In our implementation, both attention matrices—$W_{qk}$ and $W_{p}$—are row-wise normalized via softmax, thereby meeting the requirements of a probability distribution. We kindly refer the reviewer to our Response 3 to Reviewer WxDe, where we provide a more detailed explanation of the distribution of $W_{qk}$ and $W_{p}$.
>
> In contrast, using MSE loss would treat the attention weights as continuous values without accounting for their probabilistic nature. While MSE penalizes absolute deviations, it does not capture the asymmetric information divergence between two distributions. This limitation is especially critical in attention modeling, where preserving the relative probability structure across tokens is more important than minimizing element-wise differences. Thus, KL divergence provides a more principled objective by emphasizing the distributional discrepancy rather than raw magnitude differences.
>
> To further substantiate the validity of $W_{qk}$, we respectfully refer the reviewer to our Response 1 to Reviewer ohL3, in which we provide a detailed and rigorous justification.
>
> &nbsp;
>
> ---
>
> ### 6. Q7: q-k weights vs. (q-q)+(k-k) weights
>
> We respectfully note that the reason we adopt the approach of mapping object-aware query and key embeddings into pixel-aware representations to construct a more effective attention map is because self–self attention, by design, tends to allocate the majority of attention weights to the diagonal and adjacent tokens, thereby being dominated once again by local compactness. Given that we have designed a ideal attention map as the target distribution, directly computing q–k attention provides a more principled and learnable formulation than first computing q–q and k–k attentions and then aggregating them.
>
> To further validate this intuition, we performed an additional experiment on CLIP, where CAI was used to transform q and k to generate q–q + k–k attention. **The results demonstrate that our current approach achieves an average performance of 42.2, compared to only 40.8 for the q–q + k–k variant**, thereby confirming the effectiveness of our design.
>
> &nbsp;
>
> ---
>
> ### 7. Q8: Dimensions of Equation 10
>
> Thank you for pointing out this potentially confusing issue. We would like to clarify the misunderstanding. As shown in Eq. 10,    $\mathbf{S}_{\texttt{CLS}-\mathbf{T}}$
>
> is obtained by computing the cosine similarity between $\texttt{CLS}$ (with dimensions $1 \times C$) and $\mathbf{T}$ (with dimensions $C \times L$, where $L$ denotes the number of categories input to the CLIP text encoder). Similarly, $\mathbf{S}_{\mathbf{V-T}}$ is computed between $\mathbf{F_1}$ (with dimensions $\hat{H}\hat{W} \times C$) and $\mathbf{T}$. Therefore, their resulting dimensions are indeed $\mathbb{R}^L$ and $\mathbb{R}^{\hat{H}\hat{W}\times L}$, respectively.  In the revised paper, we will clarify the dimensional relationships more explicitly to prevent any possible confusion.
>
> &nbsp;
>
> We hope that our response has satisfactorily addressed your concerns, and we thank you again for your thoughtful feedback.
>
> &nbsp;
>
> Best regards,
>
> Authors

---

> ### Comment · Reviewer_ReDz · 2025-08-02
>
> Thank you for your detailed responses. Most of my concerns have been resolved. I do, however, have a follow-up question.
>
> You provided a clear explanation of the motivation behind your method and its advantages over self–self attention. However, I am curious how your method compares with CDAM [1].
>
> CDAM also does not rely on simple self–self attention. Instead, it generates attention maps using class distributions derived from the cosine similarity between image and text features. Therefore, I believe CDAM already addresses the issues you mentioned in Response W4—namely, "neglecting to maintain cross-modal alignment and disregarding the utilization of the CLIP text encoder." Moreover, CDAM operates in a training-free manner without requiring additional parameters or fine-tuning.
>
> In fact, CDAM appears to align with the same motivation you described: it leverages class distribution derived from prior models to generate new attention maps that likely preserve both local compactness and global connectivity. It is also designed to be plug-and-play compatible with CLIP-based methods.
>
> Could you please clarify how your method is meaningfully different from CDAM, particularly with respect to the motivation, architecture, or empirical behavior?
>
> [1] Kang et al., “Class Distribution-induced Attention Map for Open-vocabulary Semantic Segmentation,” ICLR 2025.

---

> > ### Author Response · Authors · 2025-08-02
> >
> > Dear reviewer,
> >
> > Thank you for your feedback. Regarding your new concern, we will review the suggested paper and attempt to integrate OPMapper into it for further experiments. We kindly ask for your patience, and we will provide you with a more detailed response shortly.
> >
> > Best regards,
> >
> > Authors

---

> ### Author Response · Authors · 2025-08-06
> **Discussion about CDAM - (1/2)**
>
> Dear reviewer ReDz:
>
> We sincerely apologize for the delay, and we are very glad to discuss with you the details of the comparision with CDAM. Over the past few days, we carefully studied the CDAM paper and its code, as you recommended for comparison. CDAM is an excellent work that ingeniously leverages the category distribution correlations in the image–text similarity map to construct attention better suited for precise localization, and it achieves strong results in experiments.
>
> That said, **our method and CDAM differ significantly in terms of motivation and innovations, model framework, adaptation process, and experimental characteristics.** In the following table, we summarize the approaches of both methods across these dimensions and highlight the key differences.
>
>
> | Dimension| OPMapper(Ours)  | CDAM(ICLR'25)|
> |---|---|---|
> | Motivation and innovations |We observe that frozen CLIP features are inherently object-aware. Under consistent value representations, **transforming the attention into pixel-aware form facilitates direct transfer to dense tasks**. The learning of the mapper, however, must **balance local compactness and global connectivity, while preserving CLIP’s inherent vision–language alignment**. | They observe that for each token (corresponding to a patch), **its classification vector distribution exhibits category dependency. Leveraging this distributional property allows them to compute more robust inter-token correlations**, ultimately yielding an attention mechanism better suited for segmentation tasks.  |
> | Architecture |A Plugin Mechanism. CAI is trained with manually designed ground-truth attention and guided by the SAA module to learn how to map q and k. **In inference, the q and k from the final CLIP layer are mapped into $q$' and $k$', which yield pixel-aware attention**. Together with the frozen $v$, these form the image features, which are then integrated with text features to produce the final mask.    |  A Rewriting Mechanism. **Using the frozen $v$ from the final CLIP layer and the enhanced text, CDAM derives a category distribution vector for each token. Attention is then computed by measuring the differences among these token-level distributions**. With this attention, $v$ is recalculated to obtain a new $v$', which is subsequently combined with the original (non-enhanced) text features to generate the mask. |
> | Image-Text Alignment | SAA performs iterative alignment of image and text features, implicitly guiding OPMapper and ultimately influencing q and k.   |CDAM lacks both explicit and implicit vision–language alignment mechanisms. As a result, the attention generated from category distribution similarity, when used together with the value features to compute $v$', may not be compatible with the frozen text features. |
> | Adaptation process | Adapt OPMapper to $q$ and $k$ of the final CLIP layer to obtain new $q$' and $k$', and then calculate image features through conventional dot-product attention with frozen $v$. | Compute similarity between text and image features, and then calculate attention by applying JS divergence to the similarity map. Finally, use the attention map and frozen $v$ to calculate the final image features. |
> | Empirical behavior | OPMapper demonstrates **more pronounced improvements overall, with especially strong performance on benchmarks with a large number of categories**: VOC-21 (+6.4), Context-60 (+5.37), Obj-171 (+6.03), Stuff-171 (+4.1), Cityscape (+8.1), and ADE-150 (+3.3). | CDAM **performs well when the number of categories is small**, but its weaker results in large-category settings reveal the limitations of relying solely on category distribution similarity: VOC-21 (+12.73), Context-60 (+4.56), Obj-171 (+4.67), Stuff-171 (+4.03), Cityscape (+3.2), and ADE-150 (+2.9). |
>
> &nbsp;
>
> **Due to space limitations, we kindly refer the reviewer to another response for more details about other differences among two methods, how to adapt OPMapper into CDAM, and the performance.**

---

> ### Author Response · Authors · 2025-08-06
> **Discussion about CDAM - (2/2)**
>
> Additional distinctions are summarized as follows:
>
> 1.	Threshold sensitivity in complex scenes: CDAM relies on a manually selected entropy-based threshold to filter category distributions. As acknowledged by the authors, this threshold is difficult to determine in complex scenes. In contrast, OPMapper requires no threshold or heuristic hyperparameter and learns attention weights in a fully end-to-end manner.
>
> 2.	Differences in global semantic modeling: CDAM enhances correlations among nearby same-category patches using JS divergence, but its handling of long-range same-category tokens still depends on thresholding. In scenarios with fragmented layouts or multiple instances, this may lead to suboptimal attention (as noted in CDAM’s limitations). OPMapper, by design, assigns high attention weights explicitly to distant same-category tokens, ensuring more consistent global connectivity.
>
> 3.	Applicability scope: CDAM is currently only applicable to training-free methods. In contrast, OPMapper is compatible with both training-free and training-based segmentation frameworks, showing broader versatility.
>
> 4.	Deviation from CLIP’s original inference mechanism: CDAM generates image features by applying the learned attention directly to the value (v) without softmax normalization, diverging from CLIP’s standard computation pipeline. OPMapper, however, solely remaps the q–k pair and retains all other aspects of CLIP’s architecture unchanged.
>
> &nbsp;
>
> By reviewing CDAM’s code, we found that it only requires reusing the q and k from the final CLIP layer after CDAM’s weight computation, feeding them into OPMapper for mapping, and then computing an additional attention weight (referred to as the OPMapper weight). The final weight is obtained as:
>
> $$
> \text{Final weight} = 0.5 \times \text{CDAM weight} + 0.5 \times \text{OPMapper weight}.
> $$
>
> The remaining steps can simply follow CDAM’s original pipeline. Therefore, our method can be seamlessly integrated with CDAM to provide further enhancements.
>
> &nbsp;
>
> We integrated OPMapper into the official CDAM implementation and evaluated it on the predefined benchmarks provided in official repository. **It is worth noting that CDAM’s attention weights do not satisfy the row-sum-to-one constraint, making their scale substantially larger than that of conventional attention. Under these circumstances, we refrain from applying any normalization to CDAM weights and simply add the OPMapper weights, which still yields clear performance gains.** The detailed results are presented in the following table:
>
> | Model | VOC-21 | Context-60 | Object-171 |
> | --- | --- | --- | --- |
> |MaskCLIP + CDAM| 55.88 |  30.49  |  34.28   |
> |MaskCLIP + CDAM + OPMapper| 56.28   | 30.87   |  34.85  |
>
> In summary, while both CDAM and OPMapper aim to produce attention weights tailored for dense prediction tasks, **they are driven by fundamentally different motivations and designs. Crucially, the two methods are complementary rather than conflicting, and can be jointly employed to further boost baseline performance.**
>
> &nbsp;
>
> **We sincerely hope that our response will gain your recognition and foster further constructive feedback.**
>
> &nbsp;
>
> Best regards,
>
> Authors

---

> ### Comment · Reviewer_ReDz · 2025-08-06
>
> I sincerely appreciate your effort in answering my questions. I have one final question: could you provide the experimental results for the mean long-range attention value comparing OPMapper and CDAM?

---

> > ### Author Response · Authors · 2025-08-06
> >
> > Dear reviewer ReDz,
> >
> > As mentioned in our earlier response, the attention weights in ODAM are not normalized with softmax. Since the mean long-range attention value is intended to reflect the extent to which attention is distributed to distant tokens, we applied normalization to the ODAM weights for this calculation. The results show that ODAM alone achieves a relatively high value—slightly lower than that of the DINO model used in ProxyCLIP—which is consistent and reasonable. We believe this performance stems from ODAM’s use of class distribution to adjust attention, further indicating that ODAM possesses a certain degree of global modeling capability.
> >
> > | Model | mean long-range attention value |
> > | -- | -- |
> > | CDAM |  0.231 |
> > | CDAM + OPMapper | 0.345 |
> > | OPMapper |  0.373 |
> >
> > When OPMapper is combined with ODAM, this value increases relative to ODAM alone, suggesting that our method introduces additional global information into the attention map. This also supports our earlier assessment of ODAM: due to the sensitivity of its threshold hyperparameter, ODAM is more effective at enhancing correlations among nearby same-category patches, while its ability to model correlations between more distant patches remains improvable. The strong global modeling ability of our method originates from our designed target attention weights, which explicitly integrate correlations among distant same-category patches during the training of OPMapper.
> >
> > In light of these results, we are pleased to conclude that achieving better integration of global and local modeling when generating attention weights for gathering information of different tokens is a promising direction for future research.
> >
> > &nbsp;
> >
> > **We truly appreciate your dedicated efforts and warmly welcome any additional constructive feedback or discussion.**
> >
> > &nbsp;
> >
> > Best regards,
> >
> > Authors

---

> ### Author Response · Authors · 2025-08-06
>
> Of coarse, we will implement the code to compute this value and follow up with the result within approximately one or two hours.

---

> ### Comment · Reviewer_ReDz · 2025-08-06
>
> I truly appreciate your efforts, and all of my concerns have been fully addressed. I will be increasing the score accordingly. It would be great if the rebuttal contents could be incorporated into the revision. If there is not enough space in the main paper, including them in the supplementary material would also be acceptable.

---

> > ### Author Response · Authors · 2025-08-06
> >
> > Dear reviewer ReDz,
> >
> > We sincerely appreciate your valuable reviews, your recognition of our response and your positive evaluation of our work. We will  follow your suggestion to include the discussion content in our revised manuscript. Wishing you joy and pleasant moments every day.
> >
> > Best regards,
> >
> > Authors

---

### Official Review · Reviewer_Jwvm · 2025-07-01

**Clarity:** 3
**Significance:** 3
**Originality:** 4
**Rating:** 5
**Confidence:** 5

**Summary:**

This paper presents OPMapper, a lightweight and effective module designed to enhance open-vocabulary semantic segmentation by improving the attention mechanisms in CLIP-based models. The authors point out that existing CLIP-based OVSS approaches either lack sufficient global semantic connectivity or demand substantial computational and data resources. OPMapper addresses these issues through two key components: Context-aware Attention Injection, which integrates both local and global semantic information into attention calculations, and Semantic Attention Alignment, which iteratively aligns attention maps with text embeddings to promote semantic coherence. The SAA module is only used during training and removed at inference time. Experiments show that this method has certain effectiveness.

**Questions:**

Please refer to the weaknesses proposed in strength and weaknesses section.

And I have some minor questions:

1. Discussion on the diminishing returns of recent training-based models on some of benchmarks. What can be the causes? How could this be mitigated in the future? This will further enhance the practicability of this work.
2. I wonder that for ProxyCLIP is the DINO encoder still kept at inference with OPMapper?
3. Are there results on the ADE-847 and PC-459 datasets?

**Ethical Concerns:**

["NO or VERY MINOR ethics concerns only"]

**Final Justification:**

After reading the authors’ response, my concerns have been resolved. I believe it is a strong paper that deserves to be published at NeurIPS. Therefore, I have maintained my positive evaluation.

**Limitations:**

Yes, authors proposed two promising directions to address their limitations.

**Quality:**

3

**Strengths And Weaknesses:**

Strengths：

1. Good motivation and easy to read. The authors pinpoint (i) the compute/annotation burden of training-based CLIP adaptations and (ii) the loss of global semantic connectivity in training-free attention patching. This crisp problem statement immediately clarifies why another method is needed and what precise gaps in the current landscape OPMapper fills.
2. Proposes a unifying, low-overhead remedy. By reframing “object-to-pixel” mapping as the sole task of a tiny adapter that re-calibrates the frozen last-layer query–key pair, the paper shows a path that eliminates heavy fine-tuning yet still injects both local compactness and global connectivity into attention maps. This makes the solution both practically attractive and conceptually elegant.
3. Solid empirical coverage. The authors graft OPMapper onto 11 distinct CLIP-based baselines (both training-free and training-based) and test on 8 public datasets.
4. Exaustive ablations. Section 5.4 and Sections in supplementary have systematically ablated each component and hyper-parameters used in this paper, demonstrating the effectiveness of its design.
5. The idea could generalise to detection, panoptic segmentation, or video.

Weaknesses：

1. Dense prose / heavy notation. Equations 3-8 introduce three attention matrices plus down-sampling without an illustrative toy example, which may cause many readers to re-derive to internalise the idea.
2. The paper directly introduces open-vocabulary semantic segmentation without first providing an overview of semantic segmentation itself. It is recommended to include a brief introduction to some SOTA semantic segmentation methods[1,2,3,4].
3. All experiments are trained on COCO-Stuff and then transferred. There is no analysis of domain shift (using different training data) or scaling to larger ViT-H/14 backbones.
4. Marginal gains on strong baselines. For the best published model (ProxyCLIP-L/14), OPMapper yields only +0.96 mIoU on average. Authors should claricy this in their rebuttal.

[1] Segmenter: Transformer for semantic segmentation. ICCV 2021.

[2] SegNeXt: Rethinking convolutional attention design for semantic segmentation. NeurIPS 2022.

[3] AUCSeg: AUC-oriented Pixel-level Long-tail Semantic Segmentation. NeurIPS 2024.

---

> ### Author Rebuttal · Authors · 2025-07-31
>
> Dear reviewer Jwvm,
>
> We sincerely thank you for your thoughtful comments, valuable insights, and the considerable time and effort you have dedicated to carefully reviewing our manuscript. The concerns and suggestions raised by you have provided us with an excellent opportunity to significantly enhance the quality and clarity of our paper. In the following, we address each concern point-by-point and present additional analyses and experiments to thoroughly resolve your concerns.
>
> &nbsp;
>
> ---
>
> ### 1. W1: Dense prose / heavy notation without an illustration
>
> We sincerely thank the reviewer for the insightful suggestion. Equations (3–8) are each associated with a specific meaning, and the three attention operations correspond to three distinct processes in our framework. In line with your suggestion, we will provide a more detailed explanation of these processes in the revised manuscript. In particular, we will add an illustrative figure to make the workflow more transparent and to ensure that readers can clearly grasp the rationale and implementation of our approach.
>
> &nbsp;
>
> ---
>
> ### 2. W2: Provide an overview of semantic segmentation for easily understanding
>
> We sincerely thank the reviewer for the valuable suggestion. In the revised version, we will incorporate an introduction to the semantic segmentation task and cite relevant related works. This addition will help provide readers with a clearer understanding of the background and motivation for open-vocabulary semantic segmentation.
>
> &nbsp;
>
> ---
>
> ### 3. W3: Scaling experiments about training data and model size
>
> We fully agree with the reviewer that the proposed scaling experiment is indeed necessary. In fact, **we have already provided results in the Table 6 of our supplementary material showing CAI trained on different datasets**, and we kindly invite the reviewer to consult the supplementary material for more details.
>
> Regarding the adaptation to larger CLIP image encoders such as ViT-H/14, we believe it will bring performance gains, though the improvement margin is expected to be narrower compared to ViT-B. In our Response 4 to Reviewer ohL3, we explained in detail why the improvement from ViT-L is smaller than that from ViT-B, and the same reasoning applies to ViT-H. Nevertheless, we sincerely appreciate this suggestion, and we plan to adapt CAI to ViT-H in our subsequent work and update the manuscript with the corresponding experimental results.
>
> Additionally, as noted in our Response 6 to Reviewer  WxDe, we have included an experiment where CAI is integrated into Cat-Seg with a ViT-L backbone, and the results further demonstrate that CAI continues to improve performance in this setting.
>
> &nbsp;
>
> ---
>
> ### 4. W4: Marginal gains on strong baselines
>
> We thank you for highlighting this interesting point, which was also observed by both you and Reviewer ReDz. First, we would like to clarify that our CAI module is extremely lightweight, and importantly, its training process and loss functions are entirely independent of the segmentation modules and categories, respectively.
>
> Then, in our experiments, we observe clear improvements in average performance among 8 benchmarks across most models. The only exception is ProxyCLIP, where the improvement is about 1%. This is because ProxyCLIP already leverages more powerful external models (e.g., DINO, MAE) to optimize its attention weights. It is consistent with the explanation we mentioned in our Response 4 to Reviewer ohL3. The parameter scale and training cost of these models are substantially larger than those of our CAI. Therefore, the relatively limited improvement (~1%) observed on ProxyCLIP is reasonable and consistent with expectations.  Importantly, **excluding ProxyCLIP, our method achieves gains ranging from 2% to 18.9% on other methods among 8 benchmarks, highlighting both the stability and the broad applicability of our approach**.
>
> &nbsp;
>
> ---
>
> ### 5. Q1: Discussion on the diminishing returns of training based methods
>
> We sincerely appreciate your valuable suggestion, which will further enhance the practicability of this work. We posit that the limited, and in some cases negative, performance improvements observed on training-based methods may stem from two primary factors. (1) Their training datasets often overlap with the benchmark categories, leading the attention to become increasingly tailored to the local–global composition favored by those methods. (2) Many of these approaches fine-tune certain backbone parameters, which in turn induces substantial changes in the distributions of q, k, and v. This limitation has also been explicitly acknowledged in our manuscript.
>
> We think (i) quantify the distributional gap between the fine-tuned and base encoders, then correct it to better accommodate OPMapper, and (ii) explicitly tie OPMapper’s parameters to the statistics of the fine-tuned model are promising directions. In the revised manuscript, we will explicitly discuss this phenomenon and make efforts to address this issue in our future work.
> &nbsp;
>
> ---
>
> ### 6. Q2: Inference details when appending CAI to ProxyCLIP
>
> Yes, the DINO encoder is still kept at inference, otherwise, the ProxyCLIP essentially degenerates into an approximate version of ClearCLIP. The outputs of CAI and DINO will be combined as the attention weights for ProxyCLIP.
>
> &nbsp;
>
> ---
>
> ### 7. Q3: Results on ADE-847 and PC-459 benchmarks
>
> We thank you for your valuable suggestion about extending experiments to more benchmarks. Typically, training-free methods do not provide results on ADE-847 and PC-459, as their performance on these large-scale open-vocabulary benchmarks is often extremely limited. Ensuring reliable performance in such large-category open-set tasks remains a challenging and valuable research direction. In our revised paper, we will make an effort to include experiments on these two benchmarks to further substantiate our approach.
>
> &nbsp;
>
> We hope that our response has satisfactorily addressed your concerns, and we thank you again for your thoughtful feedback.
>
> &nbsp;
>
> Best regards,
>
> Authors

---

> > ### Comment · Reviewer_Jwvm · 2025-08-03
> >
> > Thank you for the detailed response. I carefully read this rebuttal as well as the comments from other reviewers and the corresponding responses. I find these replies to be effective, especially as they provide additional experimental results that directly address my concerns.
> >
> > I would summarize this paper as presenting a simple yet effective approach for generating the attention masks required for dense prediction tasks. During the training of the mapping module, the method effectively addresses the lack of global connectivity often observed when other approaches rely on self-attention, while also preserving CLIP’s inherent vision–language alignment. Furthermore, the trained mapper can be applied to other methods that utilize the CLIP image encoder.
> >
> > Considering the content of this paper, its relatively high originality, broad effectiveness, and the constructive rebuttal phase with additional validation results, I am pleased to recommend the AC to accept this paper.

---

> > > ### Author Response · Authors · 2025-08-03
> > >
> > > Dear reviewer Jwvm,
> > >
> > > We sincerely appreciate your recognition of our response and your positive evaluation of our work. Wishing you joy and pleasant moments every day.
> > >
> > > Best regards,
> > >
> > > Authors

---

### Official Review · Reviewer_ohL3 · 2025-07-03

**Clarity:** 2
**Significance:** 3
**Originality:** 3
**Rating:** 4
**Confidence:** 4

**Summary:**

This paper tackles the problem of open-vocabulary semantic segmentation. It presents OPMapper, a lightweight, plug-and-play module that injects both local compactness and global connectivity into attention maps of CLIP. It introduces two core modules, Context-aware Attention Injection to embed spatial and semantic correlations, and Semantic Attention Alignment to iteratively align the enriched weights with textual prompts. The method shows promising performance on standard benchmarks.

**Questions:**

see weaknesses

**Ethical Concerns:**

["NO or VERY MINOR ethics concerns only"]

**Final Justification:**

The paper presents OPMapper, a lightweight, plug-and-play module that injects both local compactness and global connectivity into attention maps of CLIP. The method is well motivated, and the proposed module is trained-once and applied everywhere. Results are promising. The rebuttal helps clarify some key points. I appreciate it, and decide to maintain the borderline accept score.

**Limitations:**

yes

**Paper Formatting Concerns:**

did not find major formatting issues

**Quality:**

2

**Strengths And Weaknesses:**

Strengths:
+ The method is well motivated, and the proposed module is trained-once and applied-everywhere.
+ Extensive experiments are conducted, and results are promising.

Weaknesses:
- The core task in building the CAI block is to construct an ideal attention matrix. However, the proposed method is mostly based on intuitive insights, and lacks a rigorous analysis to justify why the learned attention matrix  $W_{qk}$ can be considered  ideal.
- The method introduces several hyperparameters (e.g., in Eq. (6) and Eq. (16)), but the robustness and sensitivity of these parameters are not analyzed.
- Equation (9) appears to be redundant and potentially misleading, as Equation (12) is the one actually used in the computation.
- The method demonstrates smaller overall improvements when using ViT-L/14 as the encoder compared to ViT-B/16. Please provide deeper insights regarding this result.

---

> ### Author Rebuttal · Authors · 2025-07-31
>
> Dear reviewer ohL3,
>
> We sincerely thank you for your thoughtful comments, valuable insights, and the considerable time and effort you have dedicated to carefully reviewing our manuscript. The concerns and suggestions raised by you have provided us with an excellent opportunity to significantly enhance the quality and clarity of our paper. In the following, we address each concern point-by-point and present additional analyses and experiments to thoroughly resolve your concerns.
>
> &nbsp;
>
> ---
>
> ### 1. W1: Deeper analysis of the ideal attention matrix
>
> We appreciate the reviewer for pointing out this concern regarding the theoretical rigor behind our ideal attention matrix design. We would first like to clarify that **the term 'ideal' used in our paper should be more appropriately understood as a prior attention matrix. Since it represents the target we expect the model to learn, we referred to it as 'ideal' in the current version**. However, to avoid potential confusion with an objective ideal, we will henceforth refer to it as a prior attention matrix (e.g., $W_{qk}$) in this response and update the paper accordingly.
>
> Then, we clarify that our prior attention matrix design is not a simple intuition but  fundamentally grounded on two well-established theoretical principles widely recognized in semantic segmentation literature:
>
> 1. Local Compactness: Fundamentally, semantic segmentation can be conceptualized as a problem of spatially continuous local clustering, wherein adjacent pixels are not independent but instead exhibit strong categorical coherence due to the inherent spatial smoothness of natural images. This property underscores the expectation that meaningful segmentation should leverage local contextual regularities, thereby reinforcing the robustness and consistency of the learned representations. This is commonly agreed in classical segmentation literature (FCN [Long et al., CVPR’15], DeepLab [Chen et al., TPAMI’18]). In our paper, Equation 4 explicitly incorporates Euclidean distance into attention computation to reinforce local semantic smoothness.
>
> 2. Global Connectivity: High-quality semantic segmentation fundamentally relies on the capacity to capture long-range semantic dependencies, beyond local contextual cues. This capability ensures that spatially distant pixels of the same semantic category—such as occluded regions of an identical object—are jointly recognized and consistently segmented (AAF [Ke et al., ECCV'18]). Without modeling such non-local associations, segmentation results risk fragmenting semantically coherent regions, thereby undermining both accuracy and robustness. Vision transformers (ViT, Swin Transformer) have also demonstrated this principle in modeling semantic relations between distant tokens and our Equation (5) precisely formulates a semantic-aware global connectivity pattern that aligns perfectly with this principle.
>
> To further address the reviewer’s concern with rigorous empirical evidence, We conducted the following experiment: we randomly selected 50 images containing open-set categories. For each image, we generated three attention maps using our proposed construction method, Cat-Seg (which was trained on other close-set categories but not trained on these open-set categories), and Mask2Former (which was trained on both close-set and open-set categories, and thus its attention map is treated as the "optimal" one), respectivlly. We then computed both the average cosine similarity and average Structural Similarity Index (SSIM) to measure the similarity between two attention map. As shown in the table:
>
> | Setting  |  Cosine Similarity  |  SSIM  |
> | --------- | -------------------- | ---------|
> |Ours vs Mask2Former   |     0.834      |          0.808 |
> |Cat-Seg vs Mask2Former  |     0.785    |     0.771 |
>
> **Ours (prior attention matrix), average cosine similarity = 0.834 and average SSIM = 0.808, shows a consistently high similarity compared to Cat-Seg, rigorously demonstrating that our prior attention construction is much more similar to the fully-supervised attention structure.**
>
> Furthermore, as clearly shown in Supplementary Figure 2, the attention patterns generated by CAI supervision remarkably resemble those produced by vision foundation models (e.g., DINO), providing strong qualitative validation.
>
> These analyses collectively provide rigorous justification beyond mere intuition. Therefore, we hope these can fully address the reviewer’s concern.
>
> &nbsp;
>
> ---
>
> ### 2. W2: More ablations for the choice of hyperparameters
>
> To address concerns regarding hyperparameter sensitivity and robustness, We have already conducted experiments in Tables 2 and 3 of the supplementary material for the choice of $\lambda$ and weights of different losses. We paste the results from the supplementary material below, and we kindly encourage the reviewer to refer to the supplementary material for detailed settings and explanations. The performance observed therein substantiates the rationality of our chosen hyperparameters and reinforces the reliability of the conclusions drawn in this work.
>
> > The choice of $\lambda$ in Eq.(6)
>
> |  Settings | ADE | City | Context59 | Object | VOC | Avg |
> |-----------|-------|------|-------------|---------|-------|------|
> |$\lambda$=0.1|17.8|32.5|38.2|35.9|83.8|41.6|
> |$\lambda$=0.3|18.1|32.9|38.6|36.5|84.7|42.2|
> |$\lambda$=0.5|18.0|32.6|38.1|36.0|83.7|41.7|
> |$\lambda$=0.7|17.7|32.3|38.0|36.0|83.4|41.5|
> |$\lambda$=0.9|17.0|31.7|37.3|35.6|82.6|40.8|
>
> > The choice of $w$ for different loss in Eq.(16). Settings denotes for ($w_{KL}$, $w_{DICE}$, $w_{BCE}$)
>
> |Settings | ADE  | City | Context59 | Object | VOC  | Avg. |
> |---------|------|------|-----------|--------|-------|------|
> |(5,5,5) | 16.2 | 30.7 | 36.2 | 34.5 | 81.6 |  39.8  |
> | (5,5,10) | 17.4 | 31.7 | 38.1 | 35.7 | 83.5 |  41.3  |
> | (5,10,5) | 15.7 | 30.2 | 35.4 | 34.1 | 80.9 |  39.3  |
> | (5,10,10) | 16.7 | 31.3 | 37.6 | 35.3 | 83.3 |  40.8  |
> |(10,5,5) | 17.7 | 31.9 | 38.4 | 35.9 | 83.9 |  41.6  |
> | (10,5,10)| 18.1 | 32.9 | 38.6 | 36.5 | 84.7  | 42.2 |
> | (10,10,5) | 16.5 | 31.1 | 38.0 | 35.2 | 82.6 |  40.7 |
> | (10,10,10) | 17.6 | 31.7 | 38.1 | 35.7 | 83.5 |  41.3 |
>
> &nbsp;
>
> ---
>
> ### 3. W3: Confusion about Eq 9 and Eq 12
>
> We respectfully clarify that Equation 9 and Equation 12 are not redundant. Equation 9 introduces the **general formulation** of text-guided enhancement, establishing the theoretical basis of our approach. Equation 12, on the other hand, specifies the **implementation that integrates an attention mask** which further filters out irrelative categories before merging text guidance, thereby operationalizing the general form under our proposed framework. To avoid any potential confusion, we will further clarify their relationship in our paper, ensuring that readers can clearly distinguish between the general formulation and its masked instantiation.
>
> &nbsp;
>
> ---
>
> ### 4. W4: Smaller improvements for ViT-L/14
>
> We appreciate the reviewer for raising this insightful question. The relatively smaller improvement observed with ViT-L/14 can be reasonably attributed to its inherently stronger global modeling capability due to its larger number of parameters and deeper architecture. Specifically, larger Vision Transformers (e.g., ViT-L/14) inherently exhibit more robust global connectivity, as evidenced by previous studies which consistently demonstrate improved semantic consistency and stronger long-range attention modeling as transformer models scale up in size and depth.
>
> Since our proposed OPMapper is explicitly designed to enhance both local compactness and global connectivity—particularly addressing the deficiency of global semantic connections in smaller vision transformers—the room for improvement naturally diminishes as the global modeling capability of the base encoder strengthens. **This observation is in fact a positive indication of OPMapper’s core motivation: it significantly boosts performance precisely when global connectivity is inherently limited**, such as in smaller transformer variants like ViT-B/16. Thus, the reduced improvement margin observed in ViT-L/14 experiments further validates and highlights OPMapper’s effectiveness in explicitly compensating for inadequate global connectivity.
>
> &nbsp;
>
> We hope that our response has satisfactorily addressed your concerns, and we thank you again for your thoughtful feedback.
>
> &nbsp;
>
> Best regards,
>
> Authors

---

### Author Response · Authors · 2025-08-08

Dear PC/SAC/AC and reviewers,

As the rebuttal&discussion phase draws to a close, **we would like to once again express our sincere gratitude to you for your dedicated efforts and responsible attitude throughout the review and discussion process**. The constructive suggestions provided have significantly improved the quality of our work, and we deeply appreciate them.

During the discussion period, the reviewers engaged with us in a thoughtful and detailed manner, raising insightful questions about key aspects of our approach. These concerns have inspired us to further reflect on our design and assumptions. We are also pleased that our clarifications were well received, and that the reviewers acknowledged their concerns were resolved and expressed a willingness to revise their evaluations positively.

We would also like to thank the PC/SAC/AC for their thoughtful coordination and guidance throughout the process. We are truly grateful for the opportunity to engage in such a rigorous and respectful review process, and we appreciate the professional and fair environment that has been maintained throughout.

Thank you again for your time and dedication. Wishing you joy and pleasant moments every day.

&nbsp;

Best regards,

Authors

---

### Decision · Program_Chairs · 2025-09-17

**Decision:**

Accept (poster)

**Comment:**

This paper presents work on open vocabulary semantic segmentation.  The core contribution is a module for pushing local compactness and global connectivity into a semantic segmentation method.

The reviewers initially raised concerns over the motivation and performance gains of the proposed method, while appreciating the simplicity and general applicability of the plug-in module.

After considering the authors’ responses, the reviewers converged on a general recommendation to accept the paper.  The author responses helped to clarify details of the paper and provided numerous additional pieces of empirical evidence validating the claims.

Overall, the paper provides a module for semantic segmentation that improves results and can be utilized in different paradigms for this task and is recommend for acceptance in NeurIPS.